# DM4CT: Benchmarking Diffusion Models for Computed Tomography Reconstruction

**Jiayang Shi[1,2],    Daniël M. Pelt[1],    K. Joost Batenburg[1]**
[1]LIACS, Leiden University
[2]Centrum Wiskunde en Informatica

## Abstract

Diffusion models have recently emerged as powerful priors for solving inverse problems. While Computed Tomography (CT) is theoretically a linear inverse problem, it poses many practical challenges. These include correlated noise, artifact structures, reliance on system geometry, and misaligned value ranges, which make the direct application of diffusion models more difficult than in domains like natural image generation. To systematically evaluate how diffusion models perform in this context and compare them with established reconstruction methods, we introduce DM4CT, *a comprehensive benchmark for CT reconstruction.* DM4CT includes datasets from both medical and industrial domains with sparse-view and noisy configurations. To explore the challenges of deploying diffusion models in practice, we additionally acquire a high-resolution CT dataset at a high-energy synchrotron facility and evaluate all methods under real experimental conditions. We benchmark ten recent diffusion-based methods alongside seven strong baselines, including model-based, unsupervised, and supervised approaches. Our analysis provides detailed insights into the behavior, strengths, and limitations of diffusion models for CT reconstruction. The real-world dataset is publicly available at zenodo.org/records/15420527, and the codebase is open-sourced at github.com/DM4CT/DM4CT.

## 1 Introduction

Computed Tomography (CT) is a typical example of an inverse problem, where the goal is to reconstruct an unknown object from indirect measurements (Sidky et al., 2020; Courdurier et al., 2008; Purisha et al., 2019). Techniques developed for CT reconstruction often extend to other inverse problems across various imaging applications (Clement et al., 2005; Mistretta et al., 2006; Dines & Lytle, 2005; Ladas & Devaney, 1993). In many cases, the measurements are sparse or noisy, making the reconstruction problem ill-posed. This leads to ambiguity, as multiple solutions may fit the measurements equally well. To resolve this, *prior knowledge* is typically incorporated. Approaches to utilize priors range from heuristic regularizers, such as Total Variation (TV) (Sidky & Pan, 2008; Goris et al., 2012), to learned priors via supervised deep learning (Jin et al., 2017).

Recently, diffusion models have emerged as powerful generative models, achieving remarkable success in text and image generation (Wu et al., 2023c; Li et al., 2022; Nichol et al., 2021; Ho et al., 2022). Motivated by their expressive modeling capacity, many works have proposed to use diffusion models as *learned priors* for inverse problems, showing promising results across a variety of domains (Chung et al., 2023; Song et al., 2024; Zirvi et al., 2025). Theoretically, CT reconstruction is a linear inverse problem and thus should benefit from these advances. However, practical CT imaging introduces many additional challenges. Factors such as complex noise characteristics, nonlinear preprocessing steps like log transformation, and various artifacts result in real-world CT pipelines deviating substantially from the idealized linear model (Hendriksen et al., 2020). Therefore, a comprehensive and realistic benchmark is essential to rigorously evaluate diffusion models for CT reconstruction and to compare them against other established CT reconstruction approaches.

In this work, we introduce DM4CT, a benchmark designed to evaluate diffusion-based methods for CT reconstruction. We compare diffusion models with each other and against a range of strong, established baselines. As part of DM4CT, we also propose a unified taxonomy that organizes different

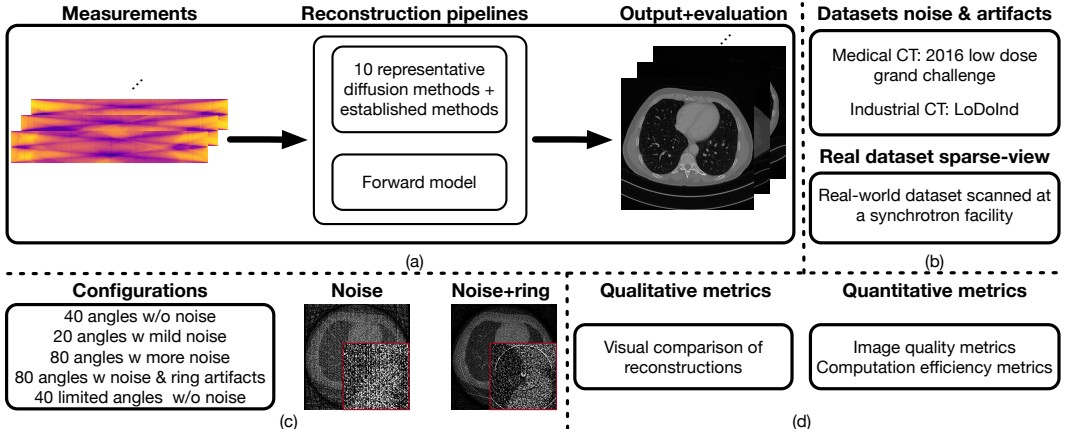

Figure 1: Overview of the DM4CT benchmark. (a) The reconstruction pipeline, where representative diffusion and baseline methods are applied to measured sinograms using the same forward model. (b) The datasets used in the benchmark, including two simulated CT datasets (medical and industrial) and one real-world dataset acquired at a synchrotron facility. (c) The five simulation configurations used to evaluate robustness to limited views, noise, and ring artifacts. Two example FBP reconstructions under noise and ring artifact conditions are shown. (d) The evaluation metrics, including both qualitative (visual) and quantitative (image quality and computational efficiency) criteria.

diffusion approaches based on their strategies for incorporating data consistency and prior knowledge (summarized in Table 1). The benchmark includes both medical and industrial CT datasets with controlled levels of noise and artifacts for objective, systematic comparison. In addition, we acquire a real-world dataset by scanning two rock samples at a synchrotron facility, allowing us to examine the limitations of deploying diffusion methods in practice under realistic conditions. An overview of our framework is illustrated in Figure 1.

Our contributions: 1) We present DM4CT, the first systematic benchmark of diffusion models for CT reconstruction; 2) We acquire and release a high-energy synchrotron CT dataset, offering a rare, well-suited resource for benchmarking under realistic conditions; 3) We propose a unified taxonomy of diffusion methods based on their strategies for data consistency and prior knowledge (Table 1); 4) We implement all benchmarked methods in the widely adopted *diffusers*[1] framework and open-source the codebase; 5) We perform extensive experiments providing practical insights into the strengths, limitations, and deployment challenges of diffusion models in CT.

*We emphasize that our goal is not to propose a new reconstruction algorithm, but to provide the first systematic benchmark for diffusion models in CT.*

## 2 PRELIMINARIES

### 2.1 COMPUTED TOMOGRAPHY

CT aims to recover an unknown object $x \in \mathbb{R}^m$ from a set of projection measurements $y \in \mathbb{R}^n$. The measurement process can be mathematically modeled as a linear system

$$y = Ax, \tag{1}$$

where $A \in \mathbb{R}^{n \times m}$ is the system matrix determined by the acquisition geometry. In practical settings, the measurements can be sparse (i.e., $n < m$), leading to an underdetermined and ill-posed inverse problem. Furthermore, the measurements acquired during scanning are typically corrupted by noise. We denote the actual observed measurements as $\widetilde{y} \in \mathbb{R}^n$. The discrepancy between the ideal and observed measurements describes the measurement noise $\epsilon = \widetilde{y} - y$. The combination of sparsity and measurement noise poses significant challenges for accurate image reconstruction in CT.

To address such challenges, prior knowledge is necessitated for the reconstruction. Classical methods often utilize heuristic priors such as TV regularization (Sidky & Pan, 2008; Goris et al., 2012; Liu

---
[1] https://github.com/huggingface/diffusers

et al., 2013; Kazantsev et al., 2018), which assume image smoothness but lack domain-specific adaptability. Recent approaches leverage data-driven priors by training deep neural networks on paired sparse- and dense-view reconstruction images (Jin et al., 2017; Chen et al., 2017; Pelt et al., 2018; Zhang et al., 2021), capturing more expressive and task-specific features. Alternatively, implicit priors such as Deep Image Prior (DIP) (Ulyanov et al., 2018; Baguer et al., 2020; Barbano et al., 2022) and Implicit Neural Representations (INRs) (Sitzmann et al., 2020; Shen et al., 2022; Wu et al., 2023b) regularize reconstruction through the neural network itself.

## 2.2 DIFFUSION MODELS

We briefly review the two types of diffusion models that are used as backbones in this work.

**Pixel Diffusion Models.** We consider Pixel-space Diffusion Probabilistic Models (DDPMs) (Ho et al., 2020; Nichol & Dhariwal, 2021) and view the forward diffusion and backward denoising processes as Stochastic Differential Equations (SDEs) (Song et al., 2020). In the forward process, Gaussian noise is gradually added to data $x_0 \sim p_{data}$ such that, at a sufficiently large time $T$, the perturbed variable $x_T$ approximates a Gaussian distribution $x_T \sim \mathcal{N}(0, I)$. This process can be described by the Variance-Preserving Stochastic Differential Equation (VP-SDE) (Song et al., 2020):

$$d\boldsymbol{x} = -\frac{\beta_t}{2}\boldsymbol{x} + \sqrt{\beta_t}d\boldsymbol{w}, \tag{2}$$

where the $\beta_t$ is a time-dependent noise schedule, and $d\boldsymbol{w}$ denotes the standard Wiener process. The backward denoising process attempts to recover samples from the original data distribution by gradually removing noise. This can be described by the corresponding reverse SDE (Anderson, 1982)

$$d\boldsymbol{x} = [-\frac{\beta_t}{2}\boldsymbol{x} - \beta_t \nabla_{\boldsymbol{x}_t} \log p(\boldsymbol{x}_t)]dt + \sqrt{\beta_t}d\bar{\boldsymbol{w}}, \tag{3}$$

where $d\bar{\boldsymbol{w}}$ is the reverse-time Wiener process, and $\nabla_{\boldsymbol{x}_t} \log p(\boldsymbol{x}_t)$ is the score function of the intermediate noisy distribution (Song & Ermon, 2019; 2020). A neural network is trained to approximate this score function, enabling sample generation by solving the reverse SDE.

**Latent Diffusion Models.** We also consider Latent Diffusion Models (LDMs) (Rombach et al., 2022), which perform the forward diffusion and reverse denoising processes in a lower-dimensional latent space instead of the pixel space. The original data $x$ is first mapped to a latent representation via an encoder $\mathcal{E}$, resulting in $z = \mathcal{E}(x)$. The forward SDE is then applied in the latent space $dz = -\frac{\beta_t}{2}z + \sqrt{\beta_t}d\boldsymbol{w}$. After the reverse denoising process in the latent domain, a decoder $\mathcal{D}$ maps the denoised latent back to the data space. In our benchmark, we use a Vector Quantized Variational Autoencoder (VQ-VAE) (Van Den Oord et al., 2017) as the encoder–decoder pair $(\mathcal{E}, \mathcal{D})$ following (Rombach et al., 2022; Rout et al., 2023; Song et al., 2024).

## 3 DM4CT

To apply diffusion models to inverse problems such as CT reconstruction, it is necessary to incorporate the measurement $y$ into the reverse denoising process. From a Bayesian perspective, the posterior distribution over the unknown image given the measurements is expressed as $p(\boldsymbol{x}|\boldsymbol{y}) \propto p(\boldsymbol{x})p(\boldsymbol{y}|\boldsymbol{x})$. This motivates a modification of the reverse-time SDE (Equation 3) to a conditional reverse SDE:

$$d\boldsymbol{x} = \left[-\frac{\beta_t}{2}\boldsymbol{x} - \beta_t \left(\nabla_{\boldsymbol{x}_t} \log p(\boldsymbol{x}_t) + \nabla_{\boldsymbol{x}_t} \log p(\boldsymbol{y}|\boldsymbol{x}_t)\right)\right] dt + \sqrt{\beta_t}\,d\bar{\boldsymbol{w}}, \tag{4}$$

where $\nabla_{\boldsymbol{x}_t} \log p(\boldsymbol{x}_t)$ is the score function approximated by the trained diffusion model, and $\nabla_{\boldsymbol{x}_t} \log p(\boldsymbol{y}|\boldsymbol{x}_t)$ introduces a measurement-informed correction. However, this measurement term is generally intractable, since $y$ typically depends on the clean image $x_0$ rather than the noised version $x_t$. A common strategy is to approximate the conditional term using the clean image estimate $\hat{x}_0(x_t)$,

$$\nabla_{\boldsymbol{x}_t} \log p(\boldsymbol{y}|\boldsymbol{x}_t) \approx \nabla_{\boldsymbol{x}_t} \log p(\boldsymbol{y}|\hat{\boldsymbol{x}}_0(\boldsymbol{x}_t)). \tag{5}$$

There are two widely used estimators for $\hat{x}_0(x_t)$, both derived from the diffusion process. The first is based on the DDPM formulation (Ho et al., 2020), $\hat{x}_0 = \frac{1}{\sqrt{\bar{\alpha}(t)}}(x_t - \sqrt{1-\bar{\alpha}(t)}\nabla_{\boldsymbol{x}_t} \log p(\boldsymbol{x}_t))$. The second uses Tweedie's formula (Song et al., 2020; Kim & Ye, 2021), which directly relates

Table 1: Diffusion-based methods evaluated in DM4CT. Columns under **Technique** refer to implementation choices (e.g., latent-space diffusion or DDIM-based sampling). Columns under **Reconstruction Strategy** denote how measurement conditioning is incorporated, including data consistency gradient steering (DC-grad), separate optimization steps (DC-step), plug-and-play priors, and use of approximate pseudoinverse solutions. A ✓* indicates only a single-step update toward the pseudoinverse. A ✓‡ indicates methods that incorporate data fidelity via a conjugate-gradient solve rather than a direct pixel-space optimization step.

| Method | Year | Technique | | Reconstruction Strategy | | | | |
|---|---|---|---|---|---|---|---|---|
| | | Latent | DDIM | DC-grad | DC-step | Plug-and-Play | Pseudo Inv | Variational Bayes |
| MCG (Chung et al., 2022) | 2022 | | | | | | ✓* | |
| DPS (Chung et al., 2023) | 2023 | | | ✓ | | | | |
| PSLD (Rout et al., 2023) | 2023 | ✓ | | ✓ | | | | |
| PGDM (Song et al., 2023a) | 2023 | | | | | | ✓ | |
| DDS (Chung et al., 2024) | 2024 | | ✓ | | ✓‡ | | | |
| Resample (Song et al., 2024) | 2024 | ✓ | ✓ | ✓ | ✓ | | | |
| DMPlug (Wang et al., 2024) | 2024 | | ✓ | | ✓ | ✓ | | |
| Reddiff (Mardani et al., 2024) | 2024 | | ✓ | ✓ | | | ✓ | ✓ |
| HybridReg (Dou et al., 2025) | 2025 | | ✓ | ✓ | | | ✓ | ✓ |
| DiffStateGrad (Zirvi et al., 2025) | 2025 | ✓ | ✓ | ✓ | ✓ | | | |

the posterior mean to the score function: $\hat{\boldsymbol{x}}_0 = \frac{1}{\sqrt{\bar{\alpha}(t)}}(\boldsymbol{x}_t - (1 - \bar{\alpha}(t))\nabla_{\boldsymbol{x}_t}\log p(\boldsymbol{x}_t))$, where $\bar{\alpha}(t) = \prod_{j=1}^{t}(1 - \beta_j)$ is the cumulative noise schedule. These approximations enable conditioning on the measurement $\boldsymbol{y}$ during the reverse denoising process in a tractable way, forming the basis for a variety of measurement-aware diffusion reconstruction methods.

## 3.1 DIFFUSION MODELS FOR CT RECONSTRUCTION

We compare nine representative diffusion-based methods for CT reconstruction. These methods differ primarily in how they incorporate the prior knowledge encoded by diffusion models into the inverse problem setting. Below, we briefly go through each strategy.

**Data Consistency Gradient.** Many methods incorporate a gradient-based data consistency steering during the reverse denoising steps (Chung et al., 2023; Rout et al., 2023; Song et al., 2024; Mardani et al., 2024; Dou et al., 2025; Zirvi et al., 2025; Tewari et al., 2023). At each timestep, after the standard denoising step, the method computes a data fidelity gradient $\boldsymbol{g}_t$ based on the estimated clean image $\hat{\boldsymbol{x}}_0(\boldsymbol{x}_t)$

$$\boldsymbol{g}_t := \nabla_{\boldsymbol{x}_t}\mathcal{L}(\boldsymbol{A}\hat{\boldsymbol{x}}_0 - \boldsymbol{y}), \tag{6}$$

where $\mathcal{L}(\cdot)$ is the loss function, e.g., $L2$ norm. This gradient is then used to steer the current iterate toward data consistency

$$\boldsymbol{x}_t \leftarrow \boldsymbol{x}_t - \eta\boldsymbol{g}_t, \tag{7}$$

with step size $\eta$ serving as a tunable hyperparameter balancing prior and data consistency. In the case of latent diffusion models, the update must propagate through the decoder $\mathcal{D}$, $\boldsymbol{g}_t := \nabla_{\boldsymbol{z}}\mathcal{L}(\boldsymbol{A}\mathcal{D}(\hat{\boldsymbol{z}}_0) - \boldsymbol{y})$   $\boldsymbol{z}_t \leftarrow \boldsymbol{z}_t - \eta\boldsymbol{g}_t$.

**Data Consistency Optimization Step.** Some methods go beyond gradient steering by inserting full data consistency optimization steps between reverse denoising iterations (Wang et al., 2024; Song et al., 2024; Zirvi et al., 2025). These steps optimize

$$\boldsymbol{x}_t^* := \arg\min_{\boldsymbol{x}_t}\mathcal{L}(\boldsymbol{A}\boldsymbol{x}_t - \boldsymbol{y}) \tag{8}$$

to directly enforce consistency to the measurements. Since this projection onto the data-consistent manifold may disrupt the reverse diffusion trajectory, some methods include a remapping step to realign $\boldsymbol{x}_t^*$ with the reverse trajectory (Song et al., 2024). In the latent setting, the step becomes $\boldsymbol{z}_t^* := \arg\min_{\boldsymbol{z}_t}\mathcal{L}(\boldsymbol{A}\mathcal{D}(\boldsymbol{z}_t) - \boldsymbol{y})$.

**Plug-and-Play.** Plug-and-play method is a powerful class of methods that incorporates prior knowledge into classical iterative reconstruction method (Venkatakrishnan et al., 2013). Unlike data

consistency gradient-based approaches, they decouple data fidelity and prior enforcement by alternating between solving a data consistency subproblem and applying an unconditional denoising step. Specifically, they optimize $\boldsymbol{x}^* = \arg\min_{\boldsymbol{x}} \mathcal{L}(\boldsymbol{A}\boldsymbol{x} - \boldsymbol{y})$, and in certain iterations, a reverse diffusion (denoising) step is applied to refine the current estimate using the learned prior (Zhu et al., 2023; Wang et al., 2024; Song et al., 2023b; Liu et al., 2023).

**Pseudo Inverse.** Several methods incorporate approximate pseudoinverse information to guide the reverse diffusion process (Chung et al., 2022; Song et al., 2023a; Mardani et al., 2024; Dou et al., 2025). Instead of relying on the data consistency gradient, these methods compute a residual between a pseudoinverse reconstruction of the measurements and that of the forward-projected image estimate

$$\boldsymbol{g}_t := \nabla_{\boldsymbol{x}_t} \mathcal{L}(\boldsymbol{A}^\dagger \boldsymbol{A}\hat{\boldsymbol{x}}_0 - \boldsymbol{A}^\dagger \boldsymbol{y}), \quad \boldsymbol{x}_t \leftarrow \boldsymbol{x}_t - \eta \boldsymbol{g}_t. \tag{9}$$

In addition, the starting noise in the reverse process may be initialized by blending random noise with the pseudoinverse reconstruction to already inject data-aware guidance at the start.

It is important to note that directly computing the Moore–Penrose inverse $\boldsymbol{A}^\dagger$ is generally infeasible in CT due to the large size and sparse structure of $\boldsymbol{A}$ (Kak & Slaney, 2001; Hansen et al., 2021; Sorzano et al., 2017). Instead, we approximate $\boldsymbol{A}^\dagger$ using Filtered BackProjection (FBP) or algebraic methods such as Simultaneous Iterative Reconstruction Technique (SIRT) (Gilbert, 1972), which serve as practical approximations to the inverse operator.

**Variational Bayesian.** In contrast to direct diffusion sampling methods that iteratively denoise an initialized noise sample, the variational Bayesian approach (Feng et al., 2023; Mardani et al., 2024; Dou et al., 2025; Dou & Song, 2024; Peng et al., 2024) approximates the posterior distribution $p(\boldsymbol{x}|\boldsymbol{y})$ using a parameterized family of distributions, typically a Gaussian. The parameters of the surrogate distribution are optimized towards both data consistency and in-distribution fit. The optimization is performed using gradient descent or similar methods, and no explicit sampling along the reverse diffusion trajectory is needed.

## 3.2 DATASETS AND CONFIGURATIONS

**Datasets.** We perform experiments on three types of CT datasets: medical, industrial, and synchrotron. The medical dataset is the 2016 Low Dose CT Grand Challenge (McCollough et al., 2017), consisting of ten patient volumes ranging from $318 \times 512 \times 512$ to $856 \times 512 \times 512$ voxels, with nine volumes used for training and one for testing. The industrial dataset, LoDoInd (Shi et al., 2024a), contains a tube filled with 15 distinct materials (e.g., coriander, pine nuts, black cumin), yielding diverse structural features and slice-wise variability. We use the central 3,500 slices of the $4000 \times 512 \times 512$ volume, with 3,000 for training and 500 for testing. In addition, we include a small case study on sparse-view reconstruction from raw medical projections in Appendix A.8.

*To highlight*, we include a high-resolution synchrotron dataset. Two rock samples of similar composition were scanned under identical conditions, resulting in reconstructed volumes of $679 \times 768 \times 768$ voxels after cropping. Compared with the medical and industrial CT datasets, our acquired synchrotron CT offers higher spatial resolution, providing fine structural details. Its simple parallel-beam, circular-trajectory geometry also enables slice-wise 2D reconstruction, substantially reducing the computational demands of benchmarking diffusion models.

For systematic evaluation, we apply controlled levels of noise and artifacts to the medical and industrial datasets, using four simulation configurations: i) 40 projection angles without noise (noise-free), ii) 20 projection angles with mild noise, iii) 80 projection angles with more noise, iv) 80 projection angles with noise and ring artifacts, v) 40 projection angles in $[0, \frac{3}{4}\pi)$ without extra noise. For the synchrotron dataset, we subsample the original 1200 projections to 200/100/60 and apply minimal preprocessing. Full details of dataset preparation are provided in Appendix A.6 and A.7. For *consistency across medical, industrial, and synchrotron datasets*, we do not use Hounsfield Unit for display, as the benchmark is *not intended for clinical analysis* but for evaluating reconstruction methods across domains.

## 3.3 IMPLEMENTATION AND COMPARISON METHODS

For each dataset, we train one pixel-space and one latent-space diffusion model, which serve as shared backbones for all diffusion-based methods to ensure fair comparison. Method-specific

Table 2: Reconstruction performance (PSNR / SSIM) of different methods under various configurations for medical, industrial and synchrotron CT datasets. The highest score among diffusion-based methods is shown in **bold**, and the second highest is underlined. A dash (–) indicates that the method exceeded the 40 GB GPU memory limit for single-slice reconstruction and is therefore not executed.

| Method | Medical | | | | | Industrial | | | | | Real-world | | |
|---|---|---|---|---|---|---|---|---|---|---|---|---|---|
| | config i | config ii | config iii | config iv | config v | config i | config ii | config iii | config iv | config v | 200 projs | 100 projs | 60 projs |
| FBP | 26.98/0.69 | 9.89/0.03 | 12.78/0.09 | 14.50/0.13 | 24.14/0.64 | 13.73/0.19 | 10.65/0.09 | 15.01/0.25 | 13.21/0.18 | 14.23/0.18 | 27.76/0.56 | 26.35/0.41 | 24.95/0.30 |
| SIRT | 30.40/0.80 | 26.23/0.47 | 24.48/0.32 | 25.86/0.40 | 26.49/0.72 | 18.40/0.38 | 16.67/0.30 | 19.17/0.40 | 17.86/0.36 | | 28.16/0.56 | 28.06/0.54 | 27.92/0.52 |
| ADMM-PDTV | 30.56/0.79 | 25.12/0.36 | 18.20/0.10 | 20.11/0.15 | 29.69/0.80 | 16.95/0.31 | 18.02/0.38 | 20.45/0.43 | 19.25/0.34 | 20.03/0.53 | 28.13/0.53 | 28.01/0.52 | 27.92/0.52 |
| FISTA-SBTV | 30.57/0.82 | 26.28/0.70 | 25.39/0.67 | 27.84/0.74 | 27.68/0.76 | 17.63/0.37 | 18.34/0.46 | 20.18/0.54 | 20.12/0.52 | 19.20/0.46 | 28.03/0.52 | 28.01/0.52 | 27.78/0.50 |
| DIP | 28.58/0.80 | 24.13/0.61 | 26.40/0.66 | 27.89/0.71 | 27.87/0.75 | 19.35/0.41 | 16.99/0.36 | 21.29/0.52 | 19.66/0.41 | 19.16/0.41 | 24.57/0.46 | 24.36/0.44 | 24.27/0.39 |
| INR | 33.21/0.86 | 26.15/0.76 | 27.74/0.80 | 29.50/0.74 | 31.01/0.81 | 20.17/0.57 | 19.44/0.52 | 22.23/0.67 | 21.41/0.60 | 20.88/0.56 | 28.01/0.50 | 28.00/0.49 | 27.92/0.48 |
| R2Gaussian | 32.14/0.81 | 24.90/0.70 | 25.45/0.74 | 25.45/0.74 | 28.26/0.78 | 18.98/0.43 | 15.98/0.22 | 18.73/0.47 | 18.87/0.49 | 17.99/0.40 | -/- | -/- | -/- |
| SwinIR | 32.45/0.88 | 29.92/0.83 | 30.37/0.84 | 30.79/0.85 | 28.93/0.81 | 22.80/0.67 | 19.51/0.55 | 25.43/0.75 | 24.84/0.74 | 22.12/0.66 | 33.75/0.76 | 33.05/0.73 | 32.41/0.70 |
| MCG | 30.00/0.79 | 27.50/0.68 | **28.90**/0.71 | 29.12/0.74 | 28.32/0.74 | 20.00/0.46 | 16.61/0.36 | **23.33**/0.59 | 21.49/0.47 | 20.38/**0.54** | 27.96/0.52 | 27.89/0.51 | 27.78/0.50 |
| DPS | 30.75/0.79 | 27.09/0.73 | 27.81/**0.74** | 28.28/0.75 | 27.47/0.72 | 21.12/0.52 | **19.40**/0.48 | 22.74/0.61 | 22.18/0.56 | 18.21/0.47 | 27.52/0.46 | 16.47/0.07 | 19.50/0.10 |
| PSLD | 26.03/0.75 | 25.12/0.73 | 25.77/**0.74** | 26.03/0.75 | 25.66/0.72 | 18.26/0.47 | 17.38/0.43 | 18.70/0.50 | 18.65/0.50 | 15.24/0.42 | 24.91/0.40 | 25.56/0.43 | 25.64/0.44 |
| PGDM | 30.26/0.80 | 27.81/0.70 | 28.44/0.66 | 29.11/0.72 | **29.66/0.77** | 21.50/0.53 | 18.92/0.41 | 23.24/**0.63** | **22.39**/0.53 | 19.33/0.49 | 27.60/0.50 | 26.31/0.46 | 26.22/0.46 |
| DDS | 31.43/0.84 | (20.12/0.22)[2] | (19.41/0.25)[2] | (18.25/0.20)[2] | 28.58/0.77 | **22.87**/0.54 | (18.23/0.39)[2] | (20.62/0.55)[2] | (19.90/0.40)[2] | **21.50**/0.51 | 28.36/0.55 | 28.10/0.51 | 27.90/0.49 |
| Resample | **32.03/0.85** | **27.92**/0.73 | 28.67/0.73 | **29.70**/0.76 | 27.70/0.76 | 18.44/0.41 | 17.32/0.34 | 19.04/0.46 | 18.46/0.32 | 16.58/0.41 | -/- | -/- | -/- |
| DMPlug | 25.77/0.71 | 25.78/0.71 | 25.81/0.71 | 25.70/0.68 | 20.70/0.50 | 18.31/0.31 | 17.87/0.33 | 18.29/0.31 | 18.57/0.31 | 18.14/0.36 | -/- | -/- | -/- |
| Reddiff | 28.01/0.78 | 26.87/0.67 | 27.66/0.70 | 27.70/0.73 | 26.04/0.73 | 20.62/**0.56** | 19.11/**0.50** | 20.73/0.50 | 21.20/0.59 | 21.40/0.53 | **28.43/0.56** | **28.24/0.54** | **28.06/0.51** |
| HybridReg | 27.63/0.78 | 26.68/0.67 | 27.40/0.71 | 27.44/0.73 | 25.74/0.73 | 20.41/0.55 | 19.00/0.50 | 20.91/0.59 | 20.86/0.58 | 21.44/**0.54** | -/- | -/- | -/- |
| DiffStateGrad | 27.46/0.77 | 26.97/**0.76** | 27.35/0.77 | 27.36/**0.77** | 24.29/0.70 | 18.47/0.39 | 19.11/**0.50** | 20.91/0.58 | 19.01/0.43 | 17.45/0.42 | -/- | -/- | -/- |

hyperparameters are tuned on held-out training subsets. We benchmark diffusion methods against a diverse set of classical and learning-based reconstruction approaches. **Classical Reconstruction.** FBP and SIRT (Gilbert, 1972), representing traditional baselines. **Deep Neural Networks as Priors.** DIP (Ulyanov et al., 2018; Baguer et al., 2020; Barbano et al., 2022), INR (Sitzmann et al., 2020; Shen et al., 2022; Wu et al., 2023b), both optimized per image without supervised training. **Gaussian Splatting–Based Reconstruction.** R2Gaussian (Zha et al., 2024), which represents objects using explicit Gaussian primitives and optimizes their parameters directly to fit CT measurements. **Model-Based Iterative Reconstruction (MBIR).** Fast Iterative Shrinkage-Thresholding Algorithm (FISTA) with Primal-Dual TV (Beck & Teboulle, 2009; Chambolle & Pock, 2011) and Alternating Direction Method of Multipliers (ADMM) with Split-Bregman TV (Boyd et al., 2011; Goldstein & Osher, 2009), enforcing structural regularity through iterative updates. **Supervised Learning.** SwinIR (Liang et al., 2021), a transformer-based image restoration model trained end-to-end to map sparse-view to dense-view reconstructions (Jin et al., 2017; Pelt et al., 2018). Full implementation and training details are provided in Appendix A.15, A.16, and A.17.

## 4 RESULTS AND DISCUSSIONS

**Reconstruction Performance.** As shown in Table 2, diffusion-based methods generally outperform classical and MBIR approaches in terms of PSNR and SSIM, but often fall short of fully supervised SwinIR. The INR-based approach achieves comparable metrics to diffusion methods, particularly in the noiseless scenario (config i) and on the real-world dataset. Visual examples in Figure 2 reveal that diffusion models tend to recover fine structural details that appear realistic but may diverge from the true reference, thereby reducing metric alignment. In contrast, INR and SwinIR produce smoother reconstructions, resulting in higher quantitative scores despite a loss of high-frequency details.

Among diffusion models, no single method or subclass (e.g., pixel vs. latent diffusion) consistently outperforms the others across all datasets and configurations, either visually or quantitatively. Performance on the real-world dataset is generally worse than on simulated data, likely due to factors such as limited training data quality and distribution shift. Perceptual metric LPIPS and full visual comparisons are discussed in Appendix A.14.

**Tradeoff between Prior and Data Consistency.** Striking the right balance between prior knowledge and data consistency is crucial for the success of diffusion-based reconstruction methods. Figure 3a illustrates this tradeoff using DPS as an example. Increasing the step size $\eta$ in Equation 7 (DC grad) initially improves both data fit and reconstruction quality. However, when $\eta$ becomes too large, the reverse denoising process is disrupted, leading to model collapse. In this regime, the reconstruction becomes dominated by measurement noise, severely degrading image quality.

---

[2]DDS is derived under an additive Gaussian noise model, solving a conjugate-gradient system of the form $(A^T A + \gamma I)$. For Poisson noise, the measurements become biased and heteroscedastic, violating the Gaussian likelihood assumption and resulting in degraded performance. See Appendix A.9 for a detailed discussion and an ablation comparing Gaussian and Poisson noise.

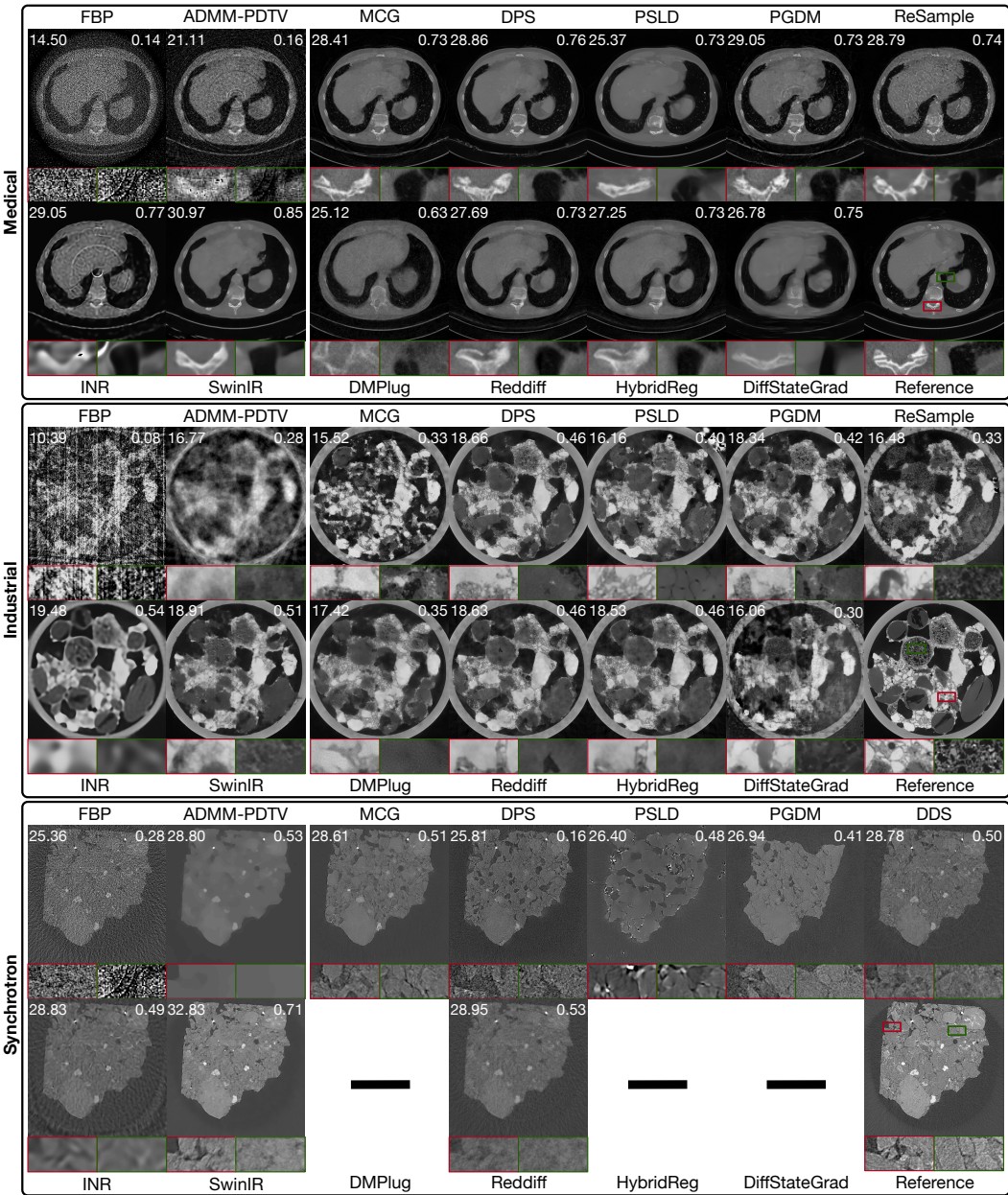

Figure 2: Reconstruction results of diffusion-based and other established methods. Top: medical dataset (config iv, 80 angles with noise & ring artifacts); middle: industrial dataset (config ii, 20 angles with mild noise); bottom: real-world synchrotron dataset (60 angles). Red and green boxes show zoom-in regions. PSNR and SSIM appear in the top-left and top-right of each image. A dash (–) indicates that the method exceeded the 40 GB GPU memory limit for single-slice reconstruction and is therefore not executed. Images are consistently linear rescaled across methods to improve contrast.

**Reconstruction Uncertainty.** Diffusion models for CT reconstruction are inherently probabilistic, enabling uncertainty quantification. Following (Antoran et al., 2023; Vasconcelos et al., 2023), we visualize the mean and standard deviation of ten MCG reconstructions from the same measurement in Figure 3b. Uncertainty is highest near structural edges—regions that are typically more ambiguous due to noise and limited-angle artifacts. Notably, outer object boundaries show high uncertainty, echoing Figure 2 and earlier observations that diffusion models struggle to capture global contours when the learned prior lacks expressiveness.

**Prior Contribution and Consistency: A Null Space Perspective.** Figure 4 illustrates how different data consistency strategies influence prior contribution, as measured by their null space components.

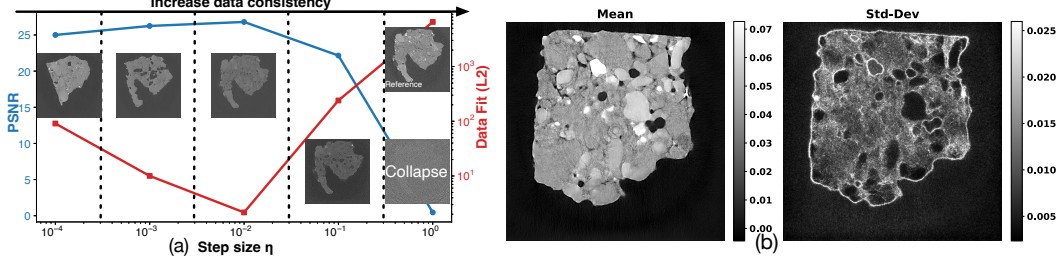

Figure 3: (a) Impact of data consistency step size $\eta$ (Equation 7) on PSNR and data fit in DPS. Moderate values improve both, while large $\eta$ disrupts denoising and causes collapse. Visual examples in the plot highlight the transition from prior-dominated to noise-dominated reconstructions. (b) Mean and standard deviation of ten MCG reconstructions conditioned on the same real measurement. Note that the real measurement used in (b) is different from the one used for (a).

DC-grad (DPS) imposes soft constraints, often allowing more content in the null space. In contrast, DC-step approaches (ReSample) enforce data consistency more strictly, resulting in smaller null components. The pseudoinverse-guided method (PGDM) offers a middle ground between those two. It highlights the trade-off between data consistency and prior-driven contents across strategies.

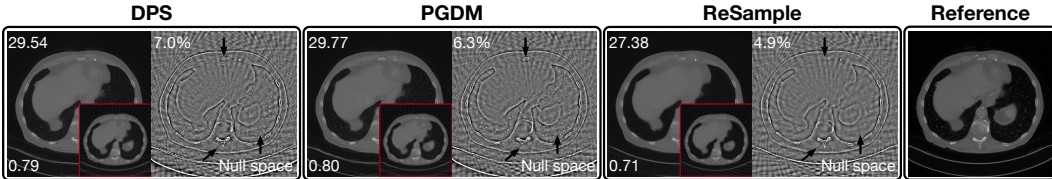

Figure 4: Decomposition of reconstructions into range and null space components for different data consistency strategies with config i). For each method, the full reconstruction is shown on the left, with zoomed-in red insets of the range component in the center and the corresponding null component on the right. The top-left of each null component indicates its relative L2 energy as a percentage of the total reconstruction, reflecting the extent of content introduced by the prior. Zoom in for details.

**Data Consistency for Latent Diffusion: Gradient or Optimization?** In latent diffusion models, enforcing data consistency via gradients is more challenging than in pixel-space diffusion (Fabian et al., 2024), as the gradients must propagate through the VQ-VAE decoder. As shown in Figure 5, PSLD (a representative latent diffusion method that relies solely on data consistency gradients) produces discontinuities in the reconstruction, even under noise-free conditions (i.e., 40 projections without noise). Similar artifacts appear frequently across configurations (see Figure 2), indicating a structural limitation of gradient-based enforcement in latent space.

In contrast, methods that incorporate explicit data consistency optimization steps, such as ReSample, can effectively correct these discontinuities and produce more coherent reconstructions in noise-free settings. However, as illustrated in Figure 5, aggressively enforcing data consistency through optimization steps can become detrimental in the presence of measurement noise (e.g., 80 projections with noise). In such cases, the reconstruction may overfit to noisy measurements, leading to degraded image quality and the amplification of noise-like features.

**Impact of Measurement Sparsity and Measurement Noise.** Figure 6 summarizes how different classes of reconstruction methods respond to variations in measurement sparsity and measurement noise. Diffusion-based methods, particularly pixel-space models, demonstrate clear advantages under sparse-view and high-noise conditions, where strong learned priors help compensate for limited or corrupted measurement information. The performance gap narrows as noise decreases or views increase, where classical and MBIR methods also become more effective.

**Computation Efficiency.** Figure 7a shows that pixel diffusion models are generally more memory- and time-efficient than latent ones. An exception is DMPlug, which uses the most memory despite being a pixel-based method. SwinIR has the fastest inference but requires substantial memory, whereas INR and DIP are memory-efficient but slower. Figure 7b details training costs. Latent diffusion involves two stages: training the VQ-VAE and then the diffusion model in latent space.

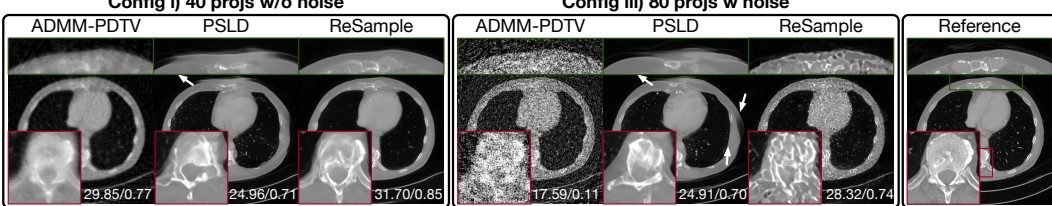

Figure 5: Reconstruction results of latent diffusion methods using only data consistency gradients (PSLD) versus additional optimization steps (ReSample) under noise-free (40 projections, no noise) and noisy (80 projections) scenarios. ADMM-PDTV serves as a classical model-based baseline that applies data consistency optimization with heuristic prior. Red insets show magnified regions.

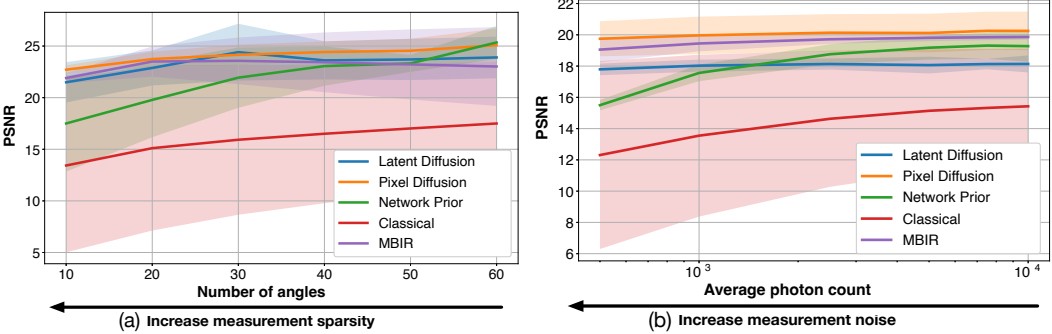

Figure 6: PSNR comparisons of method categories under (a) varying number of projection angles on the medical dataset (sparsity), and (b) different noise levels (average photon count) on the industrial dataset. Pixel diffusion refers to the six diffusion methods operating in image space; latent diffusion includes the other three latent-space methods (see Table 1). Network prior includes DIP and INR, classical methods include FBP and SIRT, and MBIR refers to ADMM-PDTV and FISTA-SBTV. Shaded regions indicate standard deviation of all methods in the category group.

Although it uses only one-third the GPU memory of pixel diffusion, the encoder alone takes as long to train as the full pixel model, making total training more costly. SwinIR demands the highest memory and training time. Ultimately, the best method depends on resource constraints and dataset size, with diffusion methods offering a flexible trade-off between training cost and inference performance.

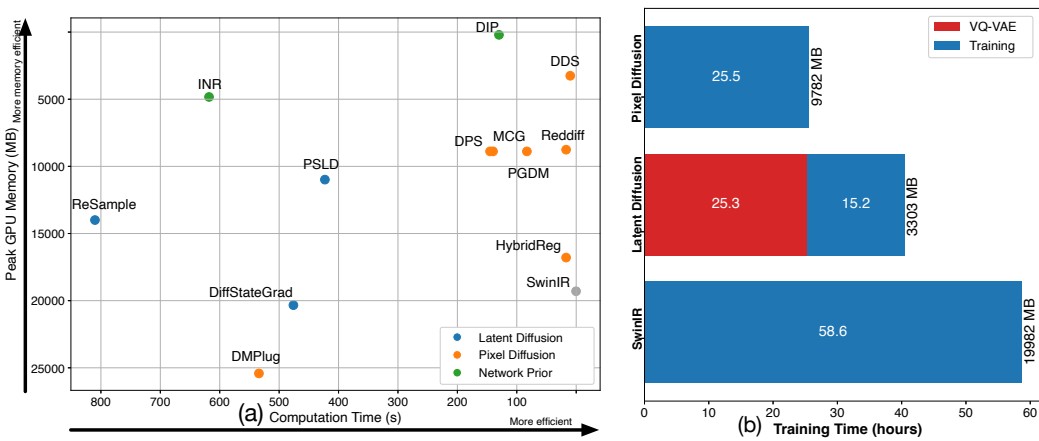

Figure 7: (a) Reconstruction time and GPU memory. The time is counted on medical dataset. (b) Training time and GPU memory of pixel diffusion, latent diffusion and SwinIR.

**Challenges in Practice.** We identify three main challenges when applying diffusion models to CT reconstruction in practice: 1) *Limited data availability:* Unlike natural images, CT datasets are often small due to privacy constraints, acquisition costs, or experimental limitations. Our synchrotron dataset narrows this gap by providing a first step toward higher-quality training data. 2) *Mismatched*

*value ranges:* CT can face inconsistent value ranges. In industrial CT, this arises from uncalibrated machines and heterogeneous materials, while in medical CT, calibration information is not always accessible, which can similarly cause misalignment. Our dataset was acquired under strictly identical conditions to mitigate this issue as much as possible, making it well-suited for benchmarking. An empirical strategy for addressing misalignment is further discussed in Appendix A.8. 3) *Computational overhead from geometry:* Diffusion methods are already resource intensive, and sophisticated geometries (e.g., helical and cone-beam) exacerbate this by requiring full 3D reconstructions. Our synchrotron dataset instead uses a straightforward parallel-beam circular trajectory, enabling efficient slice-wise 2D reconstruction and reducing geometric overhead.

## 5 CONCLUSION

We present DM4CT, a comprehensive benchmark for evaluating diffusion models for CT reconstruction. Our results demonstrate that diffusion models can serve as strong priors and achieve competitive performance across a variety of CT reconstruction scenarios. However, several key challenges remain. These include the difficulty of balancing learned priors with measurement data consistency, especially under realistic conditions involving noise, artifacts and sparse-views. While diffusion models show promise, their practical deployment for CT reconstruction is still hindered by factors as discussed. The proposed DM4CT can serve as a valuable resource for advancing future research in diffusion-based inverse problems and close the gap between methodological development and practical applicability.

**Future Work.** This benchmark highlights several promising directions for further research. First, flow-based generative models such as FlowDPS (Kim et al., 2025) are emerging as strong priors for inverse problems, and exploring their integration into CT reconstruction is an important next step. Second, combination of INRs with diffusion priors (Du et al., 2024) may offer complementary strengths, particularly for structural fidelity in sparse-view regimes. Third, while we include a preliminary downstream segmentation analysis in the appendix, a more systematic evaluation of clinical relevance (e.g., organ-level metrics, radiologist scoring) is essential for assessing practical utility. Fourth, adapting natural-image autoencoders to CT data and further examining early-stopping effects in diffusion training remain promising directions for improving training efficiency and understanding how representation quality influences reconstruction performance. Finally, a key open question is the generalizability of diffusion-based reconstruction across scanners, geometries, and acquisition protocols. Extending DM4CT with multi-institutional or cross-protocol datasets would enable rigorous testing of how well these models transfer to diverse real-world CT settings.

**Ackownledgement.** The authors acknowledge financial support by the European Union H2020-MSCA-ITN-2020 under grant agreement no. 956172 (xCTing). JS is also supported by grant from Dutch Research Council under grant no. ENWSS.2018.003 (UTOPIA) and no. NWA.1160.18.316 (CORTEX). The computation in this work is supported by SURF Snellius HPC infrastructure under grant no. EINF-15060. Synchrotron data acquisition was financially supported by the Dutch Research Council, project no. 016.Veni.192.23.

**Ethics statement.** This work adheres to the ICLR Code of Ethics. No experiments directly involve human subjects or animals. All datasets used are publicly available. The medical dataset is anonymized and complies with its original IRB protocol.

**Reproducibility statement** We have taken extensive steps to ensure reproducibility. All datasets, including our newly proposed dataset, are publicly accessible without restrictions. We open-source all code and provide detailed documentation of hyperparameter tuning ranges and procedures.

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

# A  APPENDIX

## A.1  LIMITATIONS

This work introduces DM4CT, a benchmark for evaluating diffusion-based methods in CT reconstruction. While our goal is to provide a comprehensive and systematic evaluation, several limitations remain.

First, the benchmark assumes accurate and known forward operators. In practice, system imperfections such as mechanical misalignments or calibration errors can introduce forward model inaccuracies (Ström et al., 2022; Cho et al., 2005), which are not accounted for in this study and may affect reconstruction quality in real deployments.

Second, although all diffusion methods share the same pretrained pixel and latent models as backbones, they still require method-specific hyperparameter tuning. Despite performing grid search and additional optimization, the selected hyperparameters may not be optimal for every method or scenario, especially given differing sensitivity to parameter values. This limitation also applies to the other comparison methods.

Third, while we include datasets from both medical and industrial domains, they still represent only a subset of real-world CT applications. As such, conclusions drawn from DM4CT may not fully generalize to other domains or imaging tasks.

Fourth, the evaluation relies primarily on PSNR and SSIM, which may not fully capture reconstruction fidelity in practical settings, especially when image intensity ranges vary. For example, consistent structural details with small intensity shifts can yield low scores despite visually accurate reconstructions.

Finally, while we include an exploratory downstream segmentation study, it does not guarantee performance on clinically meaningful tasks such as anatomical segmentation, organ-volume estimation, or radiologist assessment. The results depend heavily on the choice of segmentation model (SAM) and the instance-level mask-matching strategy, and therefore should be interpreted as preliminary. A more thorough downstream evaluation would be required to draw firm conclusions about the clinical applicability of diffusion-based reconstructions.

## A.2  BROADER IMPACT

This work benchmarks diffusion models in the context of CT reconstruction, a domain where accurate image recovery from incomplete or noisy measurements is critical. As generative models, diffusion approaches can effectively leverage prior knowledge to fill information gaps. However, this prior contribution may also introduce content not grounded in the measurement data, raising concerns about potential hallucination. Our benchmark aims to systematically evaluate this trade-off and promote a deeper understanding of the behavior and limitations of diffusion-based reconstruction.

In medical applications, where diagnostic decisions rely on image fidelity, rigorous clinical validation is essential before deploying such methods. In industrial settings, although the risks may differ, careful verification and domain-specific assessment are likewise important to ensure reliability and safety.

## A.3  USE OF LARGE LANGUAGE MODELS

We acknowledge the use of the large language model, ChatGPT, to assist in refining the text. The tool was utilized at the sentence level for tasks such as correcting grammar and rephrasing sentences.

## A.4  RANGE NULL SPACE DECOMPOSITION

Given a CT forward operator $A \in \mathbb{R}^{n \times m}$, it often exist at least one pseudoinverse $A^\dagger \in \mathbb{R}^{m \times n}$ satisfies $A^\dagger A = I$. Since $A A^\dagger A = A$, this implies that $A^\dagger A$ acts as a projection operator onto the range of $A$. Correspondingly, the operator $I - A^\dagger A$ projects onto the null space of $A$, as it satisfies $A(I - A^\dagger A) = 0$.

Thus, for any signal $x \in \mathbb{R}^m$, the following decomposition holds:

$$\boldsymbol{x} = \boldsymbol{A}^\dagger \boldsymbol{A} \boldsymbol{x} + (\boldsymbol{I} - \boldsymbol{A}^\dagger \boldsymbol{A}) \boldsymbol{x}, \tag{10}$$

where $\boldsymbol{A}^\dagger \boldsymbol{A} \boldsymbol{x}$ represents the *range component*, and $(\boldsymbol{I} - \boldsymbol{A}^\dagger \boldsymbol{A}) \boldsymbol{x}$ represents the *null component*.

In the context of CT reconstruction, the range component captures information directly supported by the measurements $\boldsymbol{y} = \boldsymbol{A} \boldsymbol{x}$ and reflects the data-consistent part of the solution. In contrast, the null component contains information not constrained by the measurements and instead arises from the reconstruction method, e.g., by making use of prior knowledge or learned structure. Multiple distinct null components can exist for a given measurement, giving rise to multiple feasible solutions that all satisfy the same data consistency condition.

Therefore, decomposing reconstructions into range and null space components provides a valuable perspective for analyzing how much of the reconstruction is supported by data (range) and how much is introduced by priors (null) (Wang et al.; 2023). This decomposition enables structured evaluation of data consistency and prior influence in diffusion-based CT reconstruction.

**Practical Approach for Range Null Space Decomposition.** Directly computing the pseudoinverse $\boldsymbol{A}^\dagger$ is generally infeasible in CT due to the high dimensionality and sparse structure of the system matrix $\boldsymbol{A}$ (Kak & Slaney, 2001; Hansen et al., 2021; Sorzano et al., 2017). Therefore, we adopt a more practical approach for computing the null space component without explicitly forming $\boldsymbol{A}^\dagger$. Specifically, we estimate the projection onto the null space, $(\boldsymbol{I} - \boldsymbol{A}^T \boldsymbol{A}) \boldsymbol{x}$ using an iterative method as in (Kuo et al., 2022).

We employ the Landweber iteration (Landweber, 1951), a classical algorithm for solving inverse problems, to isolate the null-space component. The procedure is outlined in Algorithm 1. Following (Kuo et al., 2022), we determine the step size $\alpha$ by first estimating the largest eigenvalue $\mu$ of the operator $\boldsymbol{A}$, and then set $\alpha = \frac{2}{0.95\mu}$ to ensure convergence while accelerating the iteration. After obtaining the null-space component $\boldsymbol{x}_{\text{null}}$, the corresponding range-space component is computed by subtraction: $\boldsymbol{x}_{\text{range}} = \boldsymbol{x} - \boldsymbol{x}_{\text{null}}$.

---

**Algorithm 1** Landweber method for computing $\boldsymbol{x}_{\text{null}}$

---

**Require:** forward operator $\boldsymbol{A}$, object $\boldsymbol{x}$, step size $\alpha$, tolerance $\varepsilon$
1: Initialize $\boldsymbol{x}^{(0)} \leftarrow \boldsymbol{x}$
2: Set $\boldsymbol{r}^{(0)} \leftarrow \boldsymbol{A}\boldsymbol{x}$
3: Set iteration number $t \leftarrow 0$
4: **while** $\|\boldsymbol{r}^{(t)}\| \geq \varepsilon$ **do**
5:     $\boldsymbol{x}^{(t+1)} \leftarrow \boldsymbol{x}^{(t)} - \alpha \boldsymbol{A}^T \boldsymbol{r}^{(t)}$
6:     $\boldsymbol{r}^{(t+1)} \leftarrow \boldsymbol{A}\boldsymbol{x}^{(t+1)}$
7:     $t \leftarrow t + 1$
8: **end while**
9: $\boldsymbol{x}_{\text{null}} \leftarrow \boldsymbol{x}^{(t)}$
10: **return** $\boldsymbol{x}_{\text{null}}$

---

### A.5 DIFFUSION MODELS FOR CT RECONSTRUCTION

Algorithm 2 outlines a common template followed by many diffusion-based methods for solving inverse problems such as CT reconstruction. These approaches typically integrate data consistency into the reverse diffusion process, guiding the sample trajectory toward agreement with the measured data. At each timestep, a clean image estimate is obtained from the noisy sample, and for example, a data consistency gradient is computed and applied to refine the current iterate. Variants of this template differ primarily in how they estimate the clean image, enforce data consistency, or combine prior and measurement information.

### A.6 EXPERIMENTAL SETUPS FOR MEDICAL AND INDUSTRIAL DATASETS

**Train/Test Dataset Split.** For the 2016 Low Dose CT Grand Challenge dataset, we use the following CT volumes for training: *L067*, *L096*, *L109*, *L143*, *L192*, *L286*, *L291*, *L310*, and *L333*. Volume *L506* is reserved for testing.

---

**Algorithm 2** Template for diffusion methods for CT reconstruction

---

**Require:** Number of diffusion steps $T$, measurement $\boldsymbol{y}$, forward operator $\boldsymbol{A}$, pretrained score function estimator $s_{\boldsymbol{\theta}}(\cdot, t)$
1:  $\boldsymbol{x}_T \sim \mathcal{N}(\mathbf{0}, \mathbf{I})$
2: **for** $t = T - 1, \cdots, 0$ **do**
3:     Estimate noise: $\hat{\boldsymbol{\epsilon}}_{t+1} \leftarrow s_{\boldsymbol{\theta}}(\boldsymbol{x}_{t+1}, t+1)$
4:     Estimate clean image: $\hat{\boldsymbol{x}}_0 \leftarrow$ through DDPM or Tweedie's formula with $\boldsymbol{x}_{t+1}$ and $\hat{\boldsymbol{\epsilon}}_{t+1}$
5:     $\boldsymbol{x}_t \leftarrow$ step backwards using $\hat{\boldsymbol{x}}_0$ and $\hat{\boldsymbol{\epsilon}}_{t+1}$ with scheduler, e.g., DDPM, DDIM
6:     $\nabla \boldsymbol{x}_t \leftarrow$ data consistency step using $\hat{\boldsymbol{x}}_0$, e.g., gradient from data consistency
7:     $\boldsymbol{x}_t \leftarrow \boldsymbol{x}_t - \eta \nabla \boldsymbol{x}_t$
8: **end for**
9: **return** Reconstructed object $\boldsymbol{x}_0$

---

For the LoDoInd dataset, which consists of 4,000 slices in total, we select slices 501–3500 for experiments. Specifically, slices 501–3000 are used for training and slices 3001–3500 are used for testing.

**Normalization.** The AAPM 2016 Low-Dose CT Grand Challenge provides HU-calibrated reconstructions. We apply a global linear mapping from the possible HU range $[\mathrm{HU}_{\min}, \mathrm{HU}_{\max}]$ of the dataset to $[-1, 1]$, ensuring consistent normalization across all volumes. For the industrial CT dataset, training and test slices originate from the same scan. We compute the global minimum and maximum over the full volume and linearly map all slices to $[-1, 1]$ using these values. All mappings are linear and invertible, original attenuation/HU values can be recovered via the inverse transform.

**Noise Simulation.** According to Beer–Lambert's law (Beer, 1852), the detected photon count $I^*$ is related to the initial photon count $I_0$, the average absorption coefficient $\gamma$, and the line integral measurement $\boldsymbol{y}_0$, as follows:

$$I^* = I_0 \exp(-\gamma \boldsymbol{y}_0), \tag{11}$$

where $\boldsymbol{y}_0$ is the clean measurement and $\gamma$ controls the attenuation strength. To simulate measurement noise, the detected photon count is modeled as a Poisson-distributed random variable:

$$\hat{I} \sim \mathrm{Poisson}(I_0 \exp(-\gamma \boldsymbol{y}_0)), \tag{12}$$

and the corresponding noisy projection is recovered by inverting the Beer–Lambert relationship:

$$\boldsymbol{y} = -\frac{1}{\gamma} \log\left(\frac{\hat{I}}{I_0}\right). \tag{13}$$

To determine the desired noise level, we first compute the average absorption of the original measurement $\boldsymbol{y}_0$ by evaluating the mean of $1 - \exp(-\boldsymbol{y}_0)$. We then adjust the intensity by scaling $\boldsymbol{y}_0$ with a constant factor $\gamma$ to match the target average absorption. For Configurations ii), iii), and iv), we set the average absorption to 50% (Table 3) by applying this scaling. The photon count $I_0$ is further varied to simulate different noise levels according to the noise model defined in Equations 12 and 13.

**Ring Artifact Simulation.** Ring artifacts arise from systematic detector defects, such as miscalibrated or malfunctioning detector elements (Sijbers & Postnov, 2004; Münch et al., 2009; Boas et al., 2012). These artifacts typically manifest as rings in the reconstructed image due to column-wise errors in the sinogram. To simulate such effects, we add fixed-pattern noise to randomly selected detector columns. Let $\boldsymbol{M}$ be a binary mask of the same shape as the measurement $\boldsymbol{y}$, where a fraction $p_{\mathrm{ring}}$ of columns are set to 1 (corrupted) and the rest to 0 (clean). Given the clean measurement $\boldsymbol{y}_0$, the corrupted measurement is generated as:

$$\boldsymbol{y} = \boldsymbol{y}_0 + \boldsymbol{M} \cdot \mathcal{N}(\mathbf{0}, \sigma^2 \boldsymbol{I}), \tag{14}$$

where $\sigma$ is the standard deviation of the Gaussian noise applied to the defective pixels.

To determine an appropriate noise level $\sigma$ for simulating ring artifacts, we first compute the standard deviation $\sigma_{\boldsymbol{y}_0}$ of the clean measurement $\boldsymbol{y}_0$. For Configuration iv) in Table 3, we set the ring artifact

intensity to $\sigma^2 = 0.25 \cdot \sigma^2_{\boldsymbol{y}_0}$, which introduces fixed-pattern perturbations to a fraction $p_{\mathrm{ring}}$ of detector columns. This simulates column-wise inconsistencies in the sinogram that manifest as ring artifacts in the reconstruction.

**Configurations**. The parameter settings for the four simulation configurations used in our benchmark are summarized in Table 3. Each configuration varies the number of projection angles and the severity of simulated noise and ring artifacts to evaluate reconstruction robustness under different levels of data corruption.

Table 3: Simulation parameters used for generating noisy and artifact-corrupted sinograms. $I_0$ controls the Poisson noise level based on the Beer–Lambert model (see Equation 11), while $p_{\mathrm{ring}}$ and $\sigma^2$ define the severity and intensity of ring artifacts (see Equation 14). $\sigma_{\boldsymbol{y}_0}$ is the standard deviation of original clean measurement $\boldsymbol{y}_0$. A value of "–" indicates no corruption of that type.

| Configuration | average absorption (noise) | $I_0$ (noise) | $p_{\mathrm{ring}}$ (ring) | $\sigma^2$ (ring) |
|---|---|---|---|---|
| i) 40 angles without noise | - | - | - | - |
| ii) 20 angles with mild noise | 50% | 10000 | - | - |
| iii) 80 angles with more noise | 50% | 5000 | - | - |
| iv) 80 angles with noise and ring artifacts | 50% | 10000 | 0.05 | $0.25 \cdot \sigma^2_{\boldsymbol{y}_0}$ |
| v) 40 angles $[0, \frac{3}{4}\pi)$ | - | - | - | - |

## A.7 REAL-WORLD SYNCHROTRON CT DATASET

We acquire a high-resolution synchrotron CT dataset at a beamline operating at 24 keV with an exposure time of 4 seconds per projection. Two rock samples are scanned using parallel-beam geometry with 1200 projections over $180°$. The detector has a pixel pitch of $9\mu m$ and the raw projection size is $679 \times 1653$ pixels. Example projection images of three different scanning angles are shown in Figure 8. To remove the background area, projections are cropped to $679 \times 768$.

Table 4: Acquisition parameters of the real-world synchrotron CT dataset used in this benchmark. Both rocks are scanned using the same setup under parallel-beam geometry.

| Sample | # Proj. | Pixel Size ($\mu$m) | Exposure (s) | Filter | Energy (keV) | # Dark Fields | # Flat Fields | Center (pre crop) | Center (post crop) | Crop Size |
|---|---|---|---|---|---|---|---|---|---|---|
| F3_1 (train) | 1200 | 9 | 4 | None | 24 | 10 | 10 | -61 | -40.5 | $679 \times 768$ |
| F3_2 (test) | 1200 | 9 | 4 | None | 24 | 10 | 10 | -62 | -41.5 | $679 \times 768$ |

**Training Reconstruction.** For the training rock, flat-field correction is performed using the median of 10 dark and 10 flat fields. Log-transformation and ring artifact reduction are applied. The ring reduction method identifies anomalous detector pixels by computing the difference between the mean and median values of each detector row (Rivers, 1998; Boin & Haibel, 2006). Full-angle FBP reconstructions are used as the training target. The reconstructions of train and test rocks using full angles, median dark/flat field and ring reduction are given in Figure 9.

**Normalization.** The two rock samples were scanned under identical settings to align value ranges as much as possible. Nevertheless, slight inter-scan mismatches remain due to noise and mild ring artifacts. After reconstruction of the training images, we prepare the test projections. We linearly rescale its projections so that the resulting reconstructions approximately match the dynamic range of the training rock.

**Benchmarking Setup.** For benchmarking, the test rock is reconstructed using only 60/100/200 evenly subsampled projections (out of 1200). A flat/dark field is randomly chosen for flat field correction and no ring artifact correction is applied.

## A.8 TOWARDS VALUE RANGE MISALIGNMENT: A SMALL CASE STUDY

A practical challenge for diffusion-based CT reconstruction is the *misalignment of value ranges*. This arises for several reasons: in industrial CT, complex material characteristics and the absence of standardized calibration make value ranges strongly dependent on scanning conditions; in medical CT, although HU are standard, the raw correction factors may be inaccessible, leading to inconsistencies;

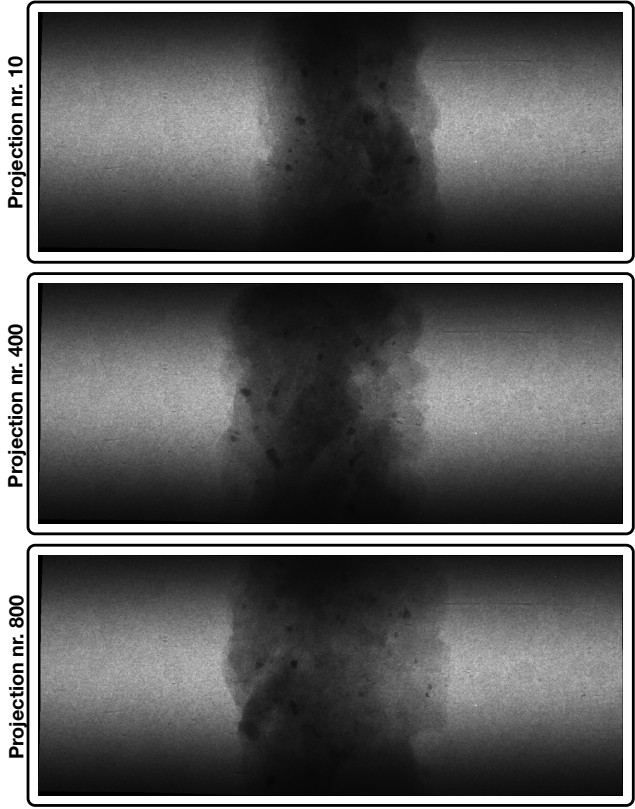

Figure 8: Example projection images (before cropping) of the synchrotron dataset at different angles.

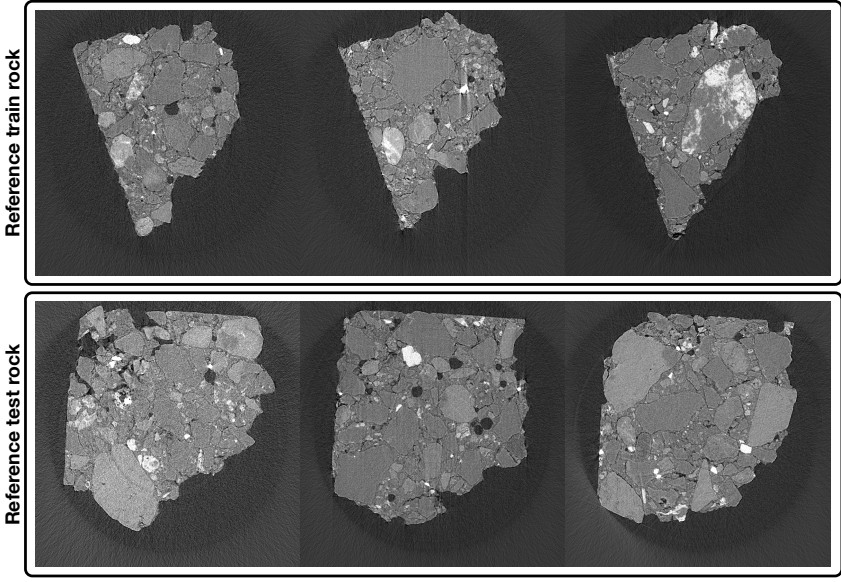

Figure 9: Reference reconstructions of three slices from the training and test rocks using all 1200 angles.

and in synchrotron CT, high photon energies and facility-dependent acquisition settings can likewise cause range shifts.

Table 5: Reconstruction from raw projections (40 angles) under value range misalignment, corrected using an empirical linear mapping. Results are approximate and intended to demonstrate feasibility rather than optimal performance. PSNR and SSIM are computed slice-wise in 2D and averaged across slices, using the corresponding physical value range of each reference slice.

| Method | FDK | SIRT | INR | MCG | DPS | DMPlug | Reddiff |
|---|---|---|---|---|---|---|---|
| PSNR/SSIM | 19.48 / 0.16 | 23.11 / 0.43 | 24.77 / 0.33 | **29.12 / 0.64** | 29.10 / 0.61 | 23.65 / 0.48 | 27.32 / 0.60 |

We present a small case study demonstrating that diffusion models can still be applied under such misalignment using a simple empirical correction. Specifically, we use diffusion models trained on the 2016 Low Dose CT Grand Challenge dataset, where training images were normalized by mapping the minimal and maximal HU values to $[-1, 1]$. For evaluation, we reconstruct from raw helical projections rebinned to fan-beam geometry following Wagner et al. (2023). As calibration factors are unavailable, the reconstructed images from raw projections do not align in value range with the normalized training images.

To address this, we select one training reconstruction and one raw-projection reconstruction at approximately corresponding anatomical locations from different patients. By estimating intensity values for background and bone regions, we establish an approximate linear mapping between the normalized training image $\boldsymbol{v}_{\text{norm}}$ and the target test image $\boldsymbol{v}_{\text{tar}}$, $\boldsymbol{v}_{\text{tar}} = a \cdot \boldsymbol{v}_{\text{norm}} + b$. This mapping allows us to sample within the normalized range $[-1, 1]$ during the diffusion reverse process, transform samples into their physical range for data consistency enforcement, and then map them back.

This procedure is purely empirical and yields only approximate reconstructions. Nevertheless, Table 5 shows results for reconstruction from raw projections with 40 angles, suggesting that such linear range alignment may serve as a practical workaround for value range misalignment when applying diffusion models in real scenarios. Figure 10 visualizes the approximate reconstructions. Despite the approximate attenuation values, the main structural features are recovered.

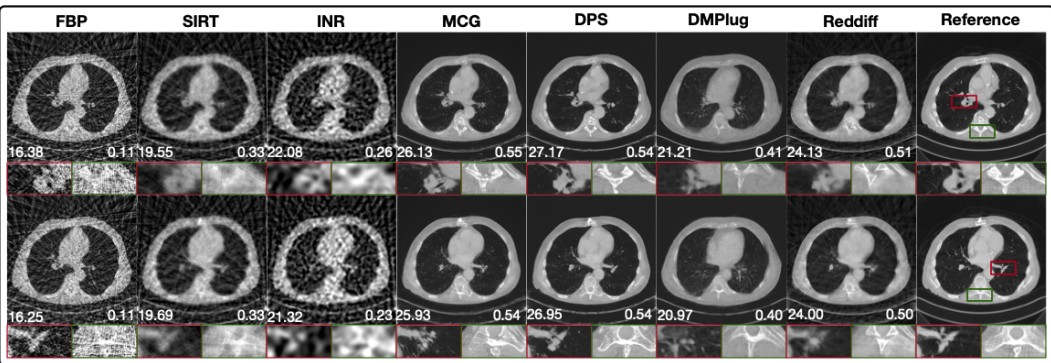

Figure 10: Reconstruction from raw projections (40 angles) under value range misalignment, corrected using an empirical linear mapping. Results are approximate and intended to demonstrate feasibility rather than optimal performance. PSNR and SSIM are shown in the lower-left and lower-right corners of each reconstruction.

## A.9 SENSITIVITY OF DDS TO NOISE MODEL

DDS (Chung et al., 2024) reconstructs from noisy measurements by solving

$$\mathcal{L}(\boldsymbol{x}) = \frac{\gamma}{2}\|\boldsymbol{y} - \boldsymbol{A}\boldsymbol{x}\|_2^2 + \frac{1}{2}\|\boldsymbol{x} - \hat{\boldsymbol{x}}_0\|_2^2,$$

where $\hat{\boldsymbol{x}}_0$ corresponds to the diffusion model's estimate of the clean signal (denoted as $\hat{\boldsymbol{x}}_t$ in the original paper). Minimizing this objective leads to the linear system

$$(\gamma\boldsymbol{A}^T\boldsymbol{A} + \boldsymbol{I})\boldsymbol{x} = \hat{\boldsymbol{x}}_0 + \gamma\boldsymbol{A}^T\boldsymbol{y},$$

which is solved via conjugate gradient. This formulation implicitly assumes a Gaussian likelihood

$$p(\boldsymbol{y} \mid \boldsymbol{x}) \propto \exp\left(-\frac{1}{2\sigma^2}\|\boldsymbol{A}\boldsymbol{x} - \boldsymbol{y}\|^2\right),$$

with noise covariance $\sigma^2 \boldsymbol{I}$.

More generally, the data-fidelity term can be written as

$$\mathcal{L}(\boldsymbol{x}) = \frac{\gamma}{2}(\boldsymbol{y} - \boldsymbol{A}\boldsymbol{x})^T \boldsymbol{R}(\boldsymbol{y} - \boldsymbol{A}\boldsymbol{x}) + \frac{1}{2}\|\boldsymbol{x} - \hat{\boldsymbol{x}}_0\|_2^2,$$

where $\boldsymbol{R} = \Sigma^{-1}$ is the inverse noise covariance. The corresponding normal equation becomes

$$(\gamma \boldsymbol{A}^T \boldsymbol{R} \boldsymbol{A} + \boldsymbol{I})\boldsymbol{x} = \hat{\boldsymbol{x}}_0 + \gamma \boldsymbol{A}^T \boldsymbol{R} \boldsymbol{y}.$$

For Gaussian noise with variance $\sigma^2$, $\boldsymbol{R} = \frac{1}{\sigma^2}\boldsymbol{I}$, and DDS reduces to tuning a single scalar hyperparameter $\sigma$.

In CT, measurements are typically corrupted by Poisson noise:

$$\sigma(y_i)^2 = \mathbb{E}[y_i] = I_0 e^{-(\boldsymbol{A}\boldsymbol{x})_i}.$$

After log transformation,

$$\mathrm{Var}(-\log(y/I_0)) \approx \frac{e^{(\boldsymbol{A}\boldsymbol{x})_i}}{I_0}$$

so the noise covariance becomes

$$\Sigma = \mathrm{diag}(\sigma_i^2), \qquad \boldsymbol{R} = \Sigma^{-1} = \mathrm{diag}\left(\frac{1}{\sigma_i^2}\right),$$

where each $\sigma_i$ depends on the forward projection $(\boldsymbol{A}\boldsymbol{x})_i$. Thus, $\boldsymbol{R}$ is *data-dependent* and cannot be absorbed into a single scalar hyperparameter $\sigma$. DDS therefore implicitly mis-specifies the likelihood under Poisson noise.

For fairness and consistency, we keep the DDS results in Table 2 using the standard Poisson noise model employed throughout our benchmark. To further illustrate the effect of noise-model mismatch, Figure 11 presents a controlled comparison in which Gaussian and Poisson noise are simulated at matched levels (i.e., producing similar FBP PSNR/SSIM). Under Gaussian noise, DDS successfully recovers fine structures, consistent with its Gaussian likelihood assumption. However, under Poisson noise (despite the same overall noise severity) the reconstruction quality degrades substantially, both visually and quantitatively. This experiment supports our observation that DDS is highly sensitive to the assumed noise model and explains its weaker performance when applied to Poisson-corrupted CT measurements.

## A.10 SEGMENTATION AS A DOWNSTREAM TASK

We use segmentation as a downstream task for the medical CT reconstructions to provide an initial exploration of how different reconstruction methods affect anatomical structure interpretation. We emphasize that DM4CT is *not* intended for clinical analysis; the goal of this subsection is to offer a preliminary technical discussion of how reconstruction quality may influence downstream tasks.

To obtain segmentation masks, we apply the Segment Anything Model (SAM) (Kirillov et al., 2023) to the reconstructed images from configuration i). Since SAM is trained on RGB images, each CT slice is duplicated across three channels before inference. The SAM-generated segmentation of the reference reconstruction is treated as the pseudo–ground truth for computing Dice (Dice, 1945) and Intersection-over-Union (IoU) (Rezatofighi et al., 2019). Because SAM may produce different discrete label sets for different reconstruction methods, we align instance masks using the commonly adopted Hungarian matching strategy (Lin et al., 2014; Kirillov et al., 2019) before evaluating metrics.

Table 6 reports the average Dice and IoU scores. Overall, diffusion-based methods underperform classical and MBIR approaches on this task. A likely explanation is that, even when classical and MBIR reconstructions are blurrier and yield lower PSNR/SSIM, they tend to preserve coarse anatomical boundaries more faithfully, resulting in greater spatial overlap with the reference segmentation.

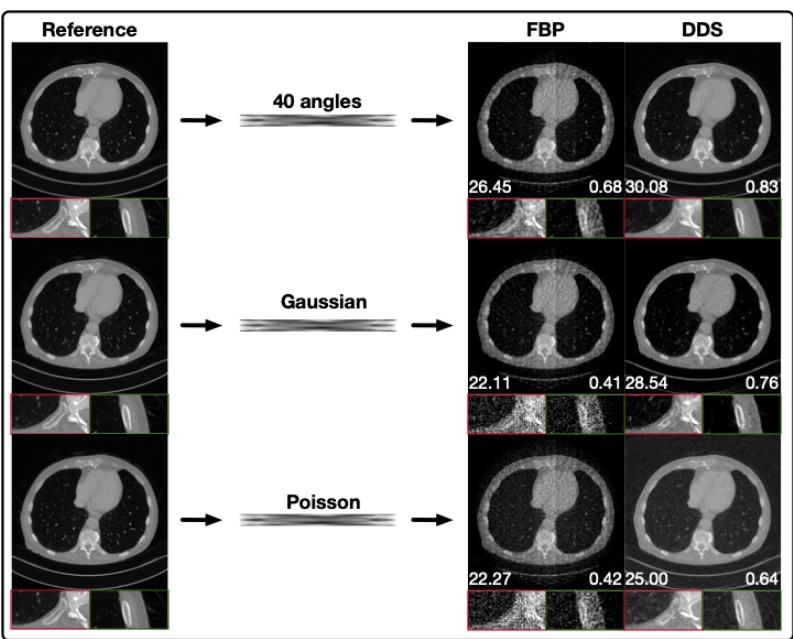

Figure 11: Effect of noise model mismatch on DDS. Gaussian and Poisson noise are simulated to produce similar FBP baselines (roughly PSNR/SSIM). DDS reconstructs fine details under Gaussian noise but degrades significantly under Poisson noise, both visually and quantitatively.

Table 6: Segmentation performance (Dice / IoU) on the medical dataset with 40 projections.

| Method | Dice / IoU |
|---|---|
| FBP | 0.603 / 0.550 |
| SIRT | 0.819 / 0.735 |
| ADMM-PDTV | 0.781 / 0.707 |
| FISTA-SBTV | 0.418 / 0.359 |
| DIP | 0.345 / 0.283 |
| INR | 0.707 / 0.633 |
| R2 Gaussian | 0.551 / 0.516 |
| SwinIR | 0.683 / 0.636 |
| MCG | 0.604 / 0.533 |
| DPS | 0.649 / 0.585 |
| PSLD | 0.381 / 0.320 |
| PGDM | 0.635 / 0.565 |
| Resample | 0.623 / 0.548 |
| DMPlug | 0.497 / 0.403 |
| Reddiff | 0.493 / 0.414 |
| DiffStateGrad | 0.385 / 0.301 |
| DDS | 0.717 / 0.657 |
| HybridReg | 0.485 / 0.405 |

In contrast, diffusion-based reconstructions often introduce subtle hallucinations or structural shifts reflecting training-set statistics rather than exact anatomy, which can reduce overlap despite producing visually cleaner images. Figure 12 visualizes the segmentation overlays and illustrates this behavior.

We stress again that this benchmark is not designed for clinical evaluation. The segmentation results here are influenced by the choice of SAM and the mask-matching protocol, and therefore serve only as an early investigation of how diffusion-based reconstructions may impact downstream tasks. More carefully controlled studies are required before drawing conclusions about clinical applicability.

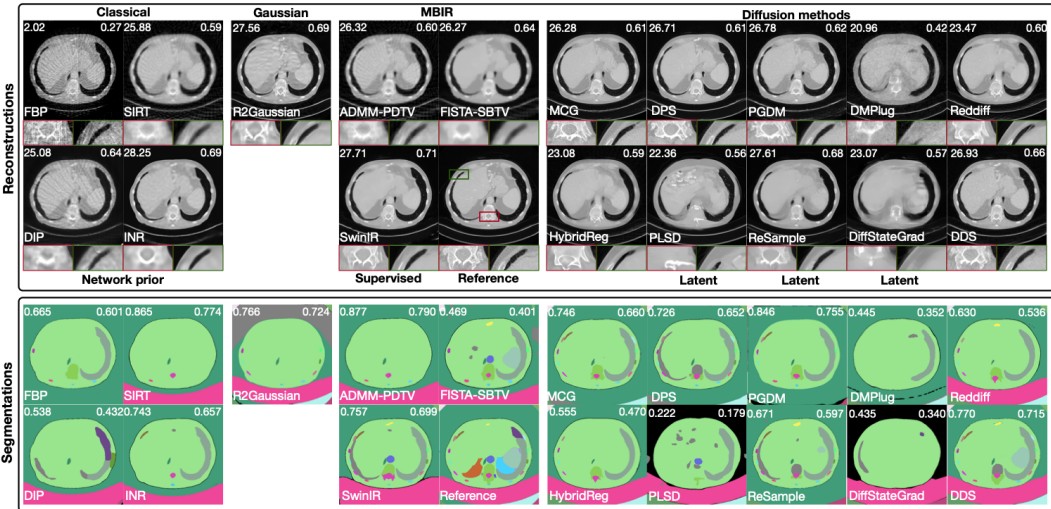

Figure 12: Segmentation masks obtained using SAM for different reconstruction methods. Dice and IoU scores are shown in the upper-left and upper-right corners of each mask, respectively. These visualizations highlight how variations in reconstruction quality influence the resulting anatomical segmentation.

### A.11   FINE TUNING EXISTING NATURAL IMAGE ENCODERS

We investigate whether natural-image autoencoders, specifically the KL-regularized variational autoencoder (AutoencoderKL) used in SDXL (Podell et al., 2023), can be adapted to CT reconstruction through fine tuning. This contrasts with the VQ-VAE used throughout our benchmark. We begin with the pretrained AutoencoderKL released by *stabilityai*[2] and evaluate two strategies: (i) using it directly without modification, and (ii) fine tuning it on CT images. We also train an AutoencoderKL (input/output channel set to 1) from scratch for comparison.

Since SDXL's AutoencoderKL is trained on RGB images, we duplicate each CT slice across three channels for both input and target. Both fine tuning and training from scratch use MSE loss with a KL regularization weight of $10^{-4}$. Fine tuning proved unstable and benefited from very small learning rates ($10^{-6}$), whereas training from scratch uses a learning rate of $10^{-4}$. After obtaining the autoencoders, we train separate latent diffusion models for each encoder for 200 epochs under identical settings.

Table 7 summarizes the representational and reconstruction capabilities of these autoencoders. The pretrained AutoencoderKL already provides reasonable representations of CT images, but fine tuning improves its autoencoder PSNR/SSIM slightly. Training AutoencoderKL from scratch unexpectedly results in lower representation quality despite using the same architecture. In contrast, the VQ-VAE used in our benchmark achieves the strongest representation quality (highest PSNR/SSIM).

These differences can stem from architectural properties. KL-regularized VAEs enforce latents to follow a standard Gaussian prior, which works well for large and diverse datasets but can degrade in low-data or low-diversity regimes (Xiao et al., 2025; Wang et al., 2021). CT datasets typically contain far fewer samples and exhibit lower structural diversity compared to natural images.

For CT reconstruction using PSLD (Rout et al., 2023), fine tuning improves the performance of AutoencoderKL compared to using it directly. The AutoencoderKL trained from scratch achieves the highest SSIM among AutoencoderKL variants, while the fine-tuned version attains slightly better PSNR. The VQ-VAE again produces the best CT reconstruction scores overall.

Figure 13 visualizes autoencoder reconstructions, unconditional diffusion samples, and CT reconstructions. The pretrained AutoencoderKL preserves fine details well in autoencoder mode but fails in conditional generation and CT reconstruction. Fine-tuned and scratch-trained models produce

---

[2]https://huggingface.co/stabilityai/sdxl-vae

Table 7: Representation (autoencoder reconstruction) and CT reconstruction quality for fine-tuned natural-image encoders. Values are PSNR/SSIM.

| Autoencoder type | config | training epochs | autoencoder reconstruction | CT reconstruction |
|---|---|---|---|---|
| AutoencoderKL (SDXL) | w/o fine tuning | 0 | 37.12/0.88 | 21.34/0.42 |
| AutoencoderKL (SDXL) | w fine tuning | 5 | 37.13/0.90 | 23.97/0.66 |
| AutoencoderKL (SDXL) | from scratch | 200 | 34.70/0.88 | 23.79/**0.72** |
| VQ-VAE (this benchmark) | from scratch | 200 | **39.30/0.92** | **25.52**/0.70 |

smoother representations and more stable reconstructions, though still lower quality than those produced by the VQ-VAE.

We emphasize that this section provides only a preliminary evaluation of fine tuning natural-image autoencoders for scientific imaging. Performance depends strongly on the specific autoencoder architecture, data scale, and diversity. CT images differ substantially from natural images in structure and statistics, and more sophisticated adaptation strategies may be required. A comprehensive investigation is therefore needed before such models can be reliably applied in CT reconstruction.

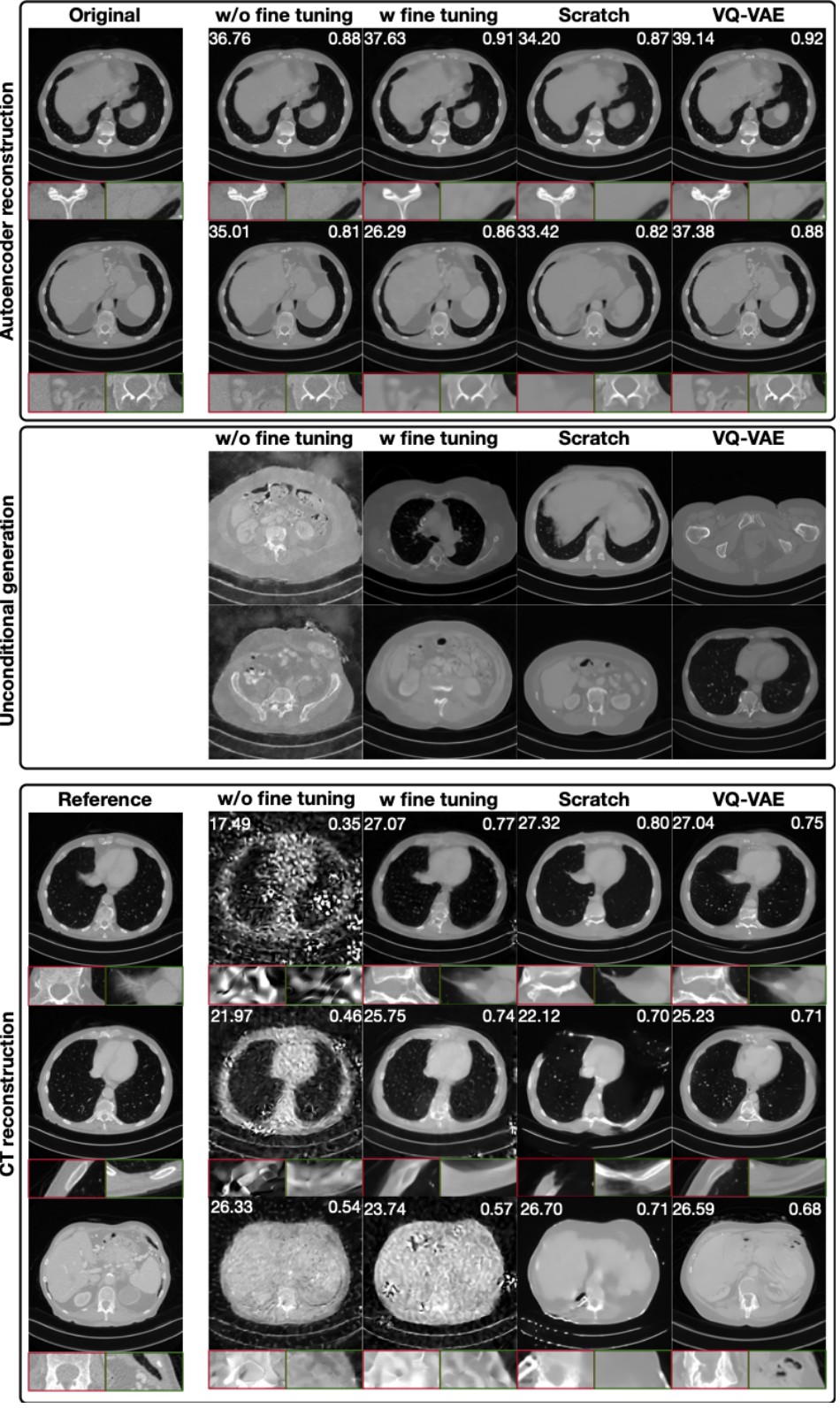

Figure 13: Comparison of autoencoder reconstruction, unconditional diffusion generation, and CT reconstruction across different autoencoders. The VQ-VAE used in our benchmark produces consistently superior representations and reconstructions, while SDXL AutoencoderKL variants exhibit reduced stability and quality.

### A.12 ABALATION FOR DATA CONSISTENCY OPTIMIZATION

We perform an ablation study to examine how the number of pixel-space and latent-space data consistency optimization iterations affects reconstruction quality under different noise conditions. In our main benchmark, these iterations are treated as fixed, and learning rates are tuned as hyperparameters. For this ablation, we instead fix both learning rates to $10^{-2}$ and vary only the number of optimization iterations.

We use the Resample method (Song et al., 2024) on the medical dataset (config iii). To study the effect of noise, we simulate measurements with identical numbers of projections but different photon statistics: a high-photon setting (10000 photons) yielding lower noise, and a low-photon setting (2500 photons) yielding higher noise.

Table 8 reports average PSNR/SSIM over ten random slices. We see that in the *high-noise* regime, increasing latent optimization iterations improves stability and yields higher PSNR/SSIM than increasing pixel iterations alone. While in the *low-noise* regime, increasing pixel iterations continues to improve reconstruction accuracy, whereas additional latent iterations provide marginal benefit.

Figure 14 visualizes selected reconstructions. When the measurement noise is high, too few iterations lead to overly smooth reconstructions lacking fine structures; conversely, too many iterations cause overfitting to noise. Interestingly, the highest PSNR/SSIM values often occur at the point where reconstructions begin to slightly overfit. This illustrates the balance between detail recovery and noise overfitting when performing optimization-based data consistency steps in diffusion pipelines.

Table 8: Average PSNR/SSIM for different combinations of pixel and latent optimization iterations under three noise conditions.

| | | Pixel iters | | | |
|---|---|---|---|---|---|
| **config iii)** | | | | | |
| | | 25 | 50 | 100 | 200 |
| Latent iters | 100 | 27.91/0.78 | 28.79/0.79 | **29.23/0.79** | 28.90/0.77 |
| | 200 | 27.90/0.78 | 28.76/0.79 | 29.20/0.79 | 28.87/0.77 |
| **config iii - more noise)** | | | | | |
| Latent iters | 100 | 27.64/0.77 | 28.40/0.77 | 28.23/0.75 | 27.77/0.72 |
| | 200 | 27.75/0.77 | 28.38/0.78 | **28.40/0.78** | 27.65/0.72 |
| **config iii - less noise)** | | | | | |
| Latent iters | 100 | 27.95/0.78 | 29.06/0.80 | 29.78/0.81 | **29.84/0.81** |
| | 200 | 27.98/0.78 | 29.04/0.80 | 29.82/0.81 | 29.82/0.81 |

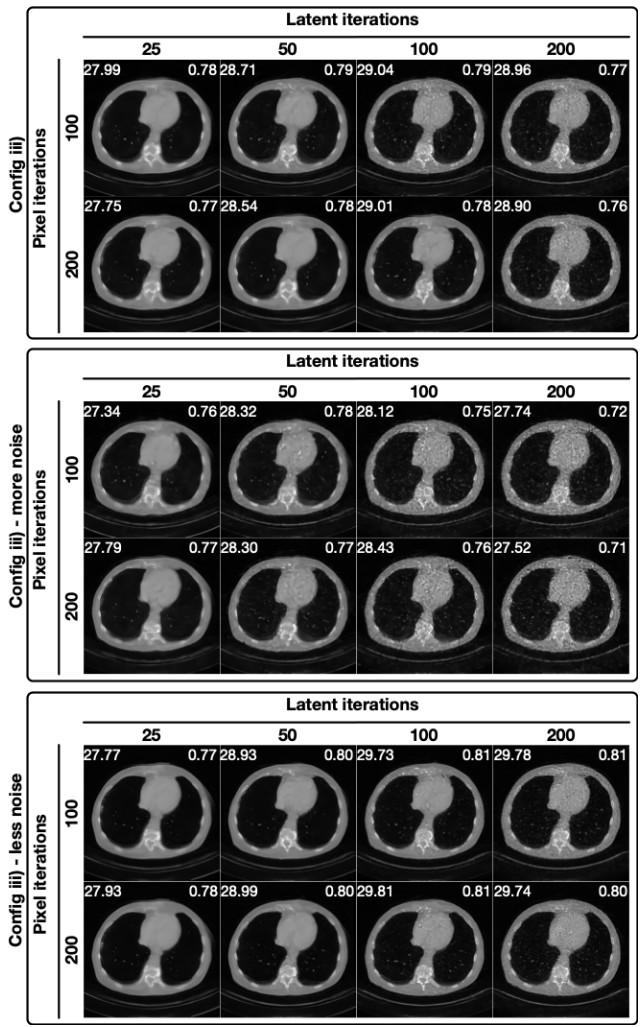

Figure 14: Effect of pixel and latent optimization iterations on reconstruction quality. Fewer iterations lead to oversmoothing, while excessive iterations cause overfitting to noise. The best-performing settings often occur near the transition between these two regimes.

## A.13 COMPARISON BETWEEN TRAINING STAGES ON RECONSTRUCTION PERFORMANCE

We investigate whether the training stage of a diffusion model affects CT reconstruction quality. Specifically, we train a standard pixel-space diffusion model for 200 epochs on the medical training dataset and save checkpoints at three stages: an early stage (25 epochs), a mid stage (100 epochs), and the final stage (200 epochs). The final-stage model is the version used throughout the benchmark.

To isolate the effect of the diffusion model itself, we evaluate these checkpoints using the DDS method (Chung et al., 2024), which does not rely on hyperparameter choices in the noiseless setting. Table 9 reports the average PSNR and SSIM on 100 randomly selected test slices.

Surprisingly, the early-stage diffusion model achieves the highest CT reconstruction accuracy by a large margin, followed by the final-stage and mid-stage models. Figure 15 visualizes unconditional samples and reconstructed CT slices for the three checkpoints. The early-stage model produces noisy unconditional generations, indicating that it has not yet learned the full data distribution. Nevertheless, when used for CT reconstruction, it yields the sharpest structures and the best fine-detail recovery. The mid-stage and final-stage models produce similar unconditional samples and reconstructions, with the final-stage model producing slightly smoother image features.

This phenomenon is intriguing and requires further investigation. Future work may explore whether this behavior generalizes to other datasets, other diffusion architectures, or other types of inverse problems. Beyond its practical implications, this finding suggests that (in some cases) early-stage diffusion models may already contain sufficiently strong structural priors for reconstruction tasks, potentially reducing diffusion training cost by up to 87.5% in inverse problem settings.

Table 9: Comparison of CT reconstruction using early stage, mid stage and final stage trained pixel diffusion models.

| Stage | Early | Mid | Final |
|---|---|---|---|
| PSNR/SSIM | **30.68/0.75** | 28.46 / 0.72 | 28.71 / 0.73 |

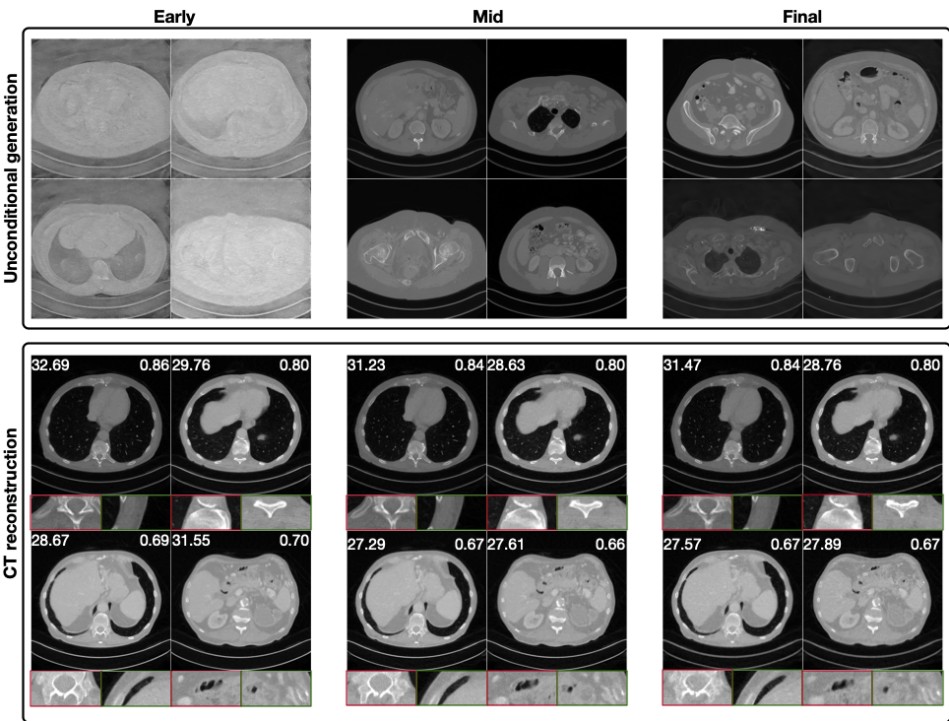

Figure 15: Visualization of unconditional generation and CT reconstruction using different stages of the trained diffusion models.

A.14   ADDITIONAL RESULTS

Figures 17, 19, and 18 present the full visual comparison of all evaluated methods across the three benchmark datasets. As observed, diffusion-based methods generally perform worse on the real-world dataset compared to the simulated datasets. This degradation may be attributed to limited high-quality training data and out-of-distribution effects. While supervised learning with SwinIR often achieves higher PSNR and SSIM scores, its reconstructions tend to be overly smooth and lack fine structural details. In contrast, diffusion models often recover more distinct structural features; however, these details can be hallucinatory and misaligned with the ground truth.

**Real-World Dataset.** Figure 17 shows that ADMM-PDTV and INR produce visually smooth reconstructions, which explains their high metrics, but they often miss finer details.

Among diffusion-based methods, DPS and PGDM reconstruct plausible object contours but introduce slight shape distortions, likely due to the *out-of-distribution* nature of the task, i.e., differences in material composition between training and test rocks. MCG better captures the overall structure but introduces unnatural porosities, hinting at a prior mismatch. PSLD fails to recover the object's correct geometry, while Reddiff struggles to preserve fine textures, likely due to the limited amount of high-quality training data. These results emphasize the increased difficulty of applying diffusion models in real CT reconstruction, where practical challenges such as noise characteristics, data scarcity, and acquisition inconsistencies must be addressed.

**Simulated Datasets.** Overall, diffusion models substantially outperform traditional methods like FBP and SIRT, particularly under realistic conditions involving noise and limited projection angles. End-to-end supervised learning with SwinIR achieves the highest scores in most configurations. In the idealized setting (configuration i: sparse views without noise), the INR method slightly outperforms diffusion-based approaches in terms of PSNR and SSIM. However, as noise and artifacts increase, diffusion methods demonstrate greater robustness and begin to outperform INR.

No single diffusion method dominates across all metrics or datasets. On the medical dataset, ReSample achieves the highest PSNR in several configurations, but its performance drops significantly on the industrial dataset, which contains more complex structures. This suggests that ReSample's strict enforcement of data consistency may work well for smooth, low-frequency features but struggles with sharper textures. Across the board, latent diffusion methods do not consistently outperform pixel-space diffusion methods.

Figure 19 and Figure 18 provide visual comparisons under challenging conditions. Classical methods like FBP and ADMM-PDTV fail to recover fine structures, while SwinIR produces smooth outputs but may lose local continuity. Diffusion models capture the global shapes well, but their reconstructions vary in detail fidelity. Interestingly, methods based purely on data consistency gradients (e.g., DPS) often yield more visually faithful results than those incorporating explicit data consistency steps (e.g., ReSample), particularly when measurements are noisy—suggesting that hard data consistency can degrade results under such conditions.

**Data Fit.** To quantify data fidelity, we compute the L2 norm between the simulated measurement $\widetilde{\boldsymbol{y}}$ and the forward projection of each reconstruction $\hat{\boldsymbol{x}}$:

$$\mathcal{L}_{\text{data\_fit}} = \|\boldsymbol{A}\hat{\boldsymbol{x}} - \widetilde{\boldsymbol{y}}\|_2. \tag{15}$$

Results across all methods and configurations are shown in Table 10. In the **noise-free settings** (config i), pixel diffusion methods (e.g., DPS, MCG) consistently achieve lower data fit loss compared to latent diffusion methods (e.g., PSLD, DMPlug), reinforcing our earlier observation that enforcing data consistency is more challenging in latent space. Pseudo-inverse guided methods such as PGDM and MCG generally achieve better data fit (lower in numbers) than purely gradient-based or plug-and-play approaches.

In **noisy configurations**, however, lower data fit does not necessarily indicate better performance: small residuals may result from overfitting to noise, while higher residuals may reflect denoising behavior. For example, although SwinIR often yields larger data fit values, its reconstructions are perceptually smoother. Thus, in noisy regimes, data fit should be interpreted alongside visual quality and robustness to overfitting.

Table 10: L2 data fit between reconstructed projections and noisy measurements ($\|A\hat{x} - \widetilde{y}\|_2$) for all benchmarked methods under different CT configurations. Lower values in noise-free settings indicate better consistency with the measurement, while in noisy cases, overly low values may reflect overfitting rather than accurate reconstruction.

| Method | Medical | | | | | Industrial | | | | | Real-world | | |
|---|---|---|---|---|---|---|---|---|---|---|---|---|---|
| | config i | config ii | config iii | config iv | config v | config i | config ii | config iii | config iv | config v | 200 projs | 100 projs | 60 projs |
| FBP | 30388.56 | 12799.31 | 12493.05 | 7770.67 | 1996.67 | 5732.75 | 6223.18 | 7191.11 | 7039.36 | 5404.93 | 534.30 | 614.72 | 808.90 |
| SIRT | 65.31 | 1178.44 | 3440.96 | 2140.96 | 297.28 | 361.80 | 389.71 | 937.56 | 878.33 | 449.79 | 244.96 | 131.53 | 71.76 |
| ADMM-PDTV | 11.79 | 1074.95 | 3025.14 | 1196.18 | 84.25 | 42.94 | 196.32 | 749.10 | 1406.61 | 161.64 | 304.55 | 226.74 | 183.23 |
| FISTA-SBTV | 209.77 | 836.49 | 2252.61 | 1680.66 | 218.40 | 588.70 | 365.39 | 1154.75 | 1515.28 | 589.53 | 307.44 | 204.25 | 145.94 |
| DIP | 659.49 | 985.68 | 2726.06 | 1802.99 | 565.09 | 677.67 | 439.45 | 1228.74 | 1714.36 | 531.00 | 1023.01 | 559.77 | 395.69 |
| INR | 213.40 | 1190.19 | 2818.56 | 1702.74 | 122.04 | 686.08 | 549.45 | 740.66 | 1265.16 | 271.24 | 341.66 | 226.68 | 165.99 |
| R2 Gaussian | 320.45 | 6347.24 | 13913.23 | 1788.97 | 460.30 | 3002.29 | 3535.71 | 5074.09 | 5131.43 | 2091.12 | -/- | -/- | -/- |
| SwinIR | 355.65 | 862.64 | 2244.76 | 1790.27 | 353.78 | 944.09 | 1313.88 | 961.93 | 1538.45 | 2125.84 | 5856.37 | 4161.74 | 3188.40 |
| MCG | 198.07 | 834.14 | 2228.57 | 1662.72 | 78.18 | 477.35 | 307.78 | 868.10 | 1354.64 | 532.25 | 410.44 | 225.70 | 159.05 |
| DPS | 261.35 | 848.95 | 2259.96 | 1792.25 | 268.79 | 768.55 | 364.61 | 1159.34 | 1488.07 | 1755.81 | 1280.33 | 4803.68 | 2878.75 |
| PSLD | 757.37 | 1064.63 | 2476.15 | 2058.66 | 892.60 | 2822.79 | 1789.62 | 4158.79 | 4409.87 | 4312.18 | 3791.18 | 1996.16 | 1387.63 |
| PGDM | 209.98 | 826.91 | 2307.33 | 1664.57 | 83.20 | 338.59 | 311.40 | 774.78 | 1247.13 | 1143.05 | 464.36 | 1081.48 | 847.48 |
| DDS | 14.91 | 3979.23 | 8164.71 | 15382.82 | 18.06 | 15.36 | 2455.27 | 4919.95 | 2828.38 | 26.76 | 189.96 | 91.42 | 4100.51 |
| Resample | 736.59 | 846.29 | 2272.55 | 1775.01 | 976.97 | 4367.23 | 1988.47 | 6962.81 | 7276.90 | 6036.66 | -/- | -/- | -/- |
| DMPlug | 941.97 | 1020.57 | 2547.04 | 2169.60 | 2497.78 | 1705.36 | 2528.21 | 2541.57 | 1748.81 | | -/- | -/- | -/- |
| Reddiff | 360.21 | 857.07 | 2270.13 | 1695.80 | 196.03 | 681.36 | 388.39 | 1287.98 | 1610.73 | 8091.12 | 188.48 | 95.12 | 52.21 |
| HybridReg | 419.04 | 862.45 | 2286.82 | 1722.13 | 230.88 | 781.52 | 439.16 | 1410.80 | 1706.94 | 79.27 | -/- | -/- | -/- |
| DiffStateGrad | 951.74 | 979.61 | 2503.64 | 2110.76 | 1089.06 | 3289.29 | 1734.75 | 4765.78 | 4848.37 | 50.76 | -/- | -/- | -/- |

Table 11: LPIPS for all benchmarked methods under different CT configurations. Lower values mean better perceptual alignment.

| Method | Medical | | | | | Industrial | | | | | Real-world | | |
|---|---|---|---|---|---|---|---|---|---|---|---|---|---|
| | config i | config ii | config iii | config iv | config v | config i | config ii | config iii | config iv | config v | 200 projs | 100 projs | 60 projs |
| FBP | 0.59 | 0.57 | 1.09 | 1.04 | 0.37 | 0.65 | 0.76 | 0.63 | 0.78 | 0.63 | 0.20 | 0.30 | 0.37 |
| SIRT | 0.41 | 1.16 | 0.64 | 0.57 | 0.35 | 0.64 | 0.73 | 0.53 | 0.56 | 0.63 | 0.42 | 0.40 | 0.41 |
| ADMM-PDTV | 0.38 | 0.63 | 0.93 | 0.84 | 0.25 | 0.62 | 0.70 | 0.51 | 0.56 | 0.40 | 0.46 | 0.49 | 0.50 |
| FISTA-SBTV | 0.40 | 0.52 | 0.52 | 0.46 | 0.40 | 0.72 | 0.75 | 0.66 | 0.66 | 0.72 | 0.56 | 0.56 | 0.57 |
| DIP | 0.38 | 0.50 | 0.46 | 0.40 | 0.32 | 0.50 | 0.59 | 0.41 | 0.46 | 0.50 | 0.39 | 0.39 | 0.41 |
| INR | 0.30 | 0.52 | 0.46 | 0.43 | 0.29 | 0.63 | 0.59 | 0.41 | 0.45 | 0.48 | 0.55 | 0.55 | 0.56 |
| R2 Gaussian | 0.25 | 0.32 | 0.37 | 0.42 | 0.32 | 0.67 | 0.67 | 0.55 | 0.54 | 0.60 | -/- | -/- | -/- |
| SwinIR | 0.20 | 0.33 | 0.28 | 0.26 | 0.19 | 0.24 | 0.35 | 0.20 | 0.24 | 0.22 | 0.27 | 0.30 | 0.27 |
| MCG | 0.14 | 0.30 | 0.27 | 0.22 | 0.11 | 0.29 | 0.37 | 0.19 | 0.28 | 0.27 | 0.25 | 0.24 | 0.24 |
| DPS | 0.13 | 0.21 | 0.19 | 0.18 | 0.13 | 0.24 | 0.28 | 0.18 | 0.21 | 0.32 | 0.24 | 0.39 | 0.38 |
| PSLD | 0.29 | 0.31 | 0.29 | 0.28 | 0.24 | 0.38 | 0.40 | 0.36 | 0.36 | 0.46 | 0.42 | 0.39 | 0.38 |
| PGDM | 0.16 | 0.29 | 0.36 | 0.27 | 0.12 | 0.24 | 0.34 | 0.18 | 0.24 | 0.34 | 0.26 | 0.37 | 0.38 |
| DDS | 0.09 | 0.61 | 0.91 | 0.84 | 0.10 | 0.21 | 0.35 | 0.26 | 0.39 | 0.28 | 0.16 | 0.16 | 0.17 |
| Resample | 0.22 | 0.34 | 0.34 | 0.31 | 0.26 | 0.42 | 0.48 | 0.37 | 0.41 | 0.44 | -/- | -/- | -/- |
| DMPlug | 0.30 | 0.30 | 0.30 | 0.32 | 0.42 | 0.41 | 0.41 | 0.42 | 0.41 | 0.40 | -/- | -/- | -/- |
| Reddiff | 0.28 | 0.32 | 0.29 | 0.26 | 0.27 | 0.31 | 0.34 | 0.29 | 0.28 | 0.34 | 0.29 | 0.33 | 0.36 |
| HybridReg | 0.30 | 0.32 | 0.29 | 0.26 | 0.28 | 0.33 | 0.35 | 0.31 | 0.29 | 0.34 | -/- | -/- | -/- |
| DiffStateGrad | 0.35 | 0.36 | 0.35 | 0.35 | 0.35 | 0.44 | 0.48 | 0.42 | 0.42 | 0.44 | -/- | -/- | -/- |

**Perceptual Metrics.** In addition to PSNR and SSIM, we evaluate reconstructions using the perceptual metric LPIPS (Zhang et al., 2018), reported in Table 11. While reconstruction accuracy is typically the primary goal in CT, perceptual metrics can provide complementary insights into how well reconstructions align with human perception. To compute LPIPS, we duplicate each grayscale CT reconstruction across three channels and use AlexNet as the backbone, following standard practice. As Table 11 shows, LPIPS scores for individual methods do not always correlate with PSNR/SSIM (Table 2), highlighting that perceptual similarity may capture different aspects of reconstruction quality. Nevertheless, the overall trends are consistent with our earlier findings: diffusion-based methods generally achieve comparable or slightly worse perceptual scores than the supervised SwinIR baseline, but clearly outperform conventional approaches such as FBP and SIRT.

**Data Consistency Strategies in Null Space Perspective.** Figure 16 presents range–null space decompositions for DPS, PGDM, MCG, and ReSample under configuration iv) of the industrial dataset, which includes both noise and ring artifacts. These methods implement distinct strategies for enforcing data consistency.

Consistent with findings in Section 4, DPS (which uses data consistency gradients) imposes only soft constraints, resulting in a substantial null space component indicative of prior contributed features. PGDM, which incorporates a pseudoinverse guidance, shows better alignment with the measurement, yielding a lower null energy. Interestingly, MCG, which applies only a single step toward pseudoinverse, exhibits higher null energy than DPS, suggesting that a single-step update may not sufficiently enforce consistency.

ReSample, which uses explicit data consistency optimization steps, achieves the lowest null space energy, reflecting strong enforcement. However, unlike in the noiseless case, such strict enforcement in this noisy and artifact-prone scenario leads to degraded visual quality. Despite suppressing ring artifacts, ReSample introduces new structured distortions, possibly due to the optimization process redistributing ring-related inconsistencies across the reconstruction in an effort to maintain global consistency.

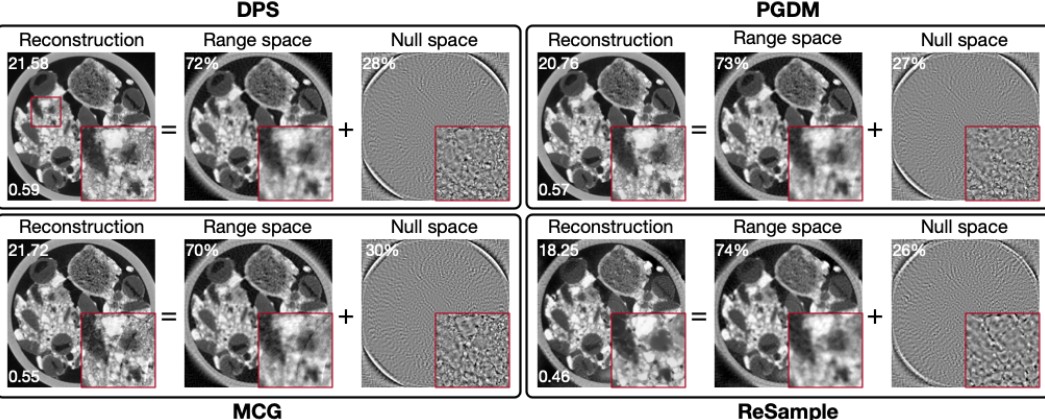

Figure 16: Range-null space decompositions of four reconstruction methods with different data consistency strategies, evaluated on the industrial dataset (config iv: 80 projections with noise and ring artifacts). PSNR and SSIM are shown in the top-left and bottom-left corners of the reconstruction images. Energy percentages of the range and null space components are shown in their respective top-left corners.

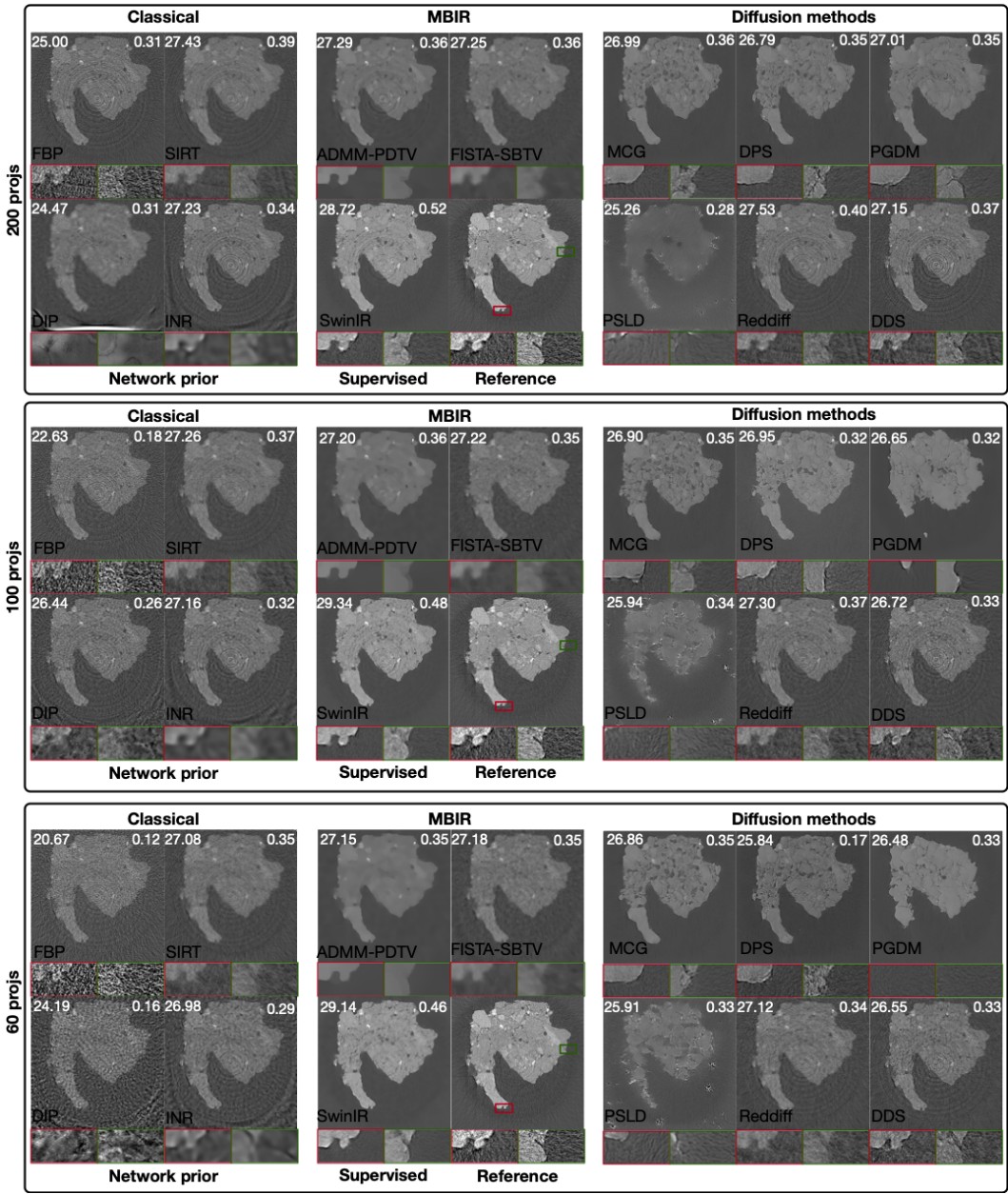

Figure 17: Visual comparison of all benchmarked methods on the real-world synchrotron CT dataset under three different sparse-view scenarios (200, 100, and 60 projections). PSNR and SSIM values are shown in the top-left and top-right corners of each image, respectively. Red and green boxes mark zoom-in regions for structural detail comparison. The reference image is reconstructed using all 1200 projections, median flat-field correction, and additional ring artifact suppression.

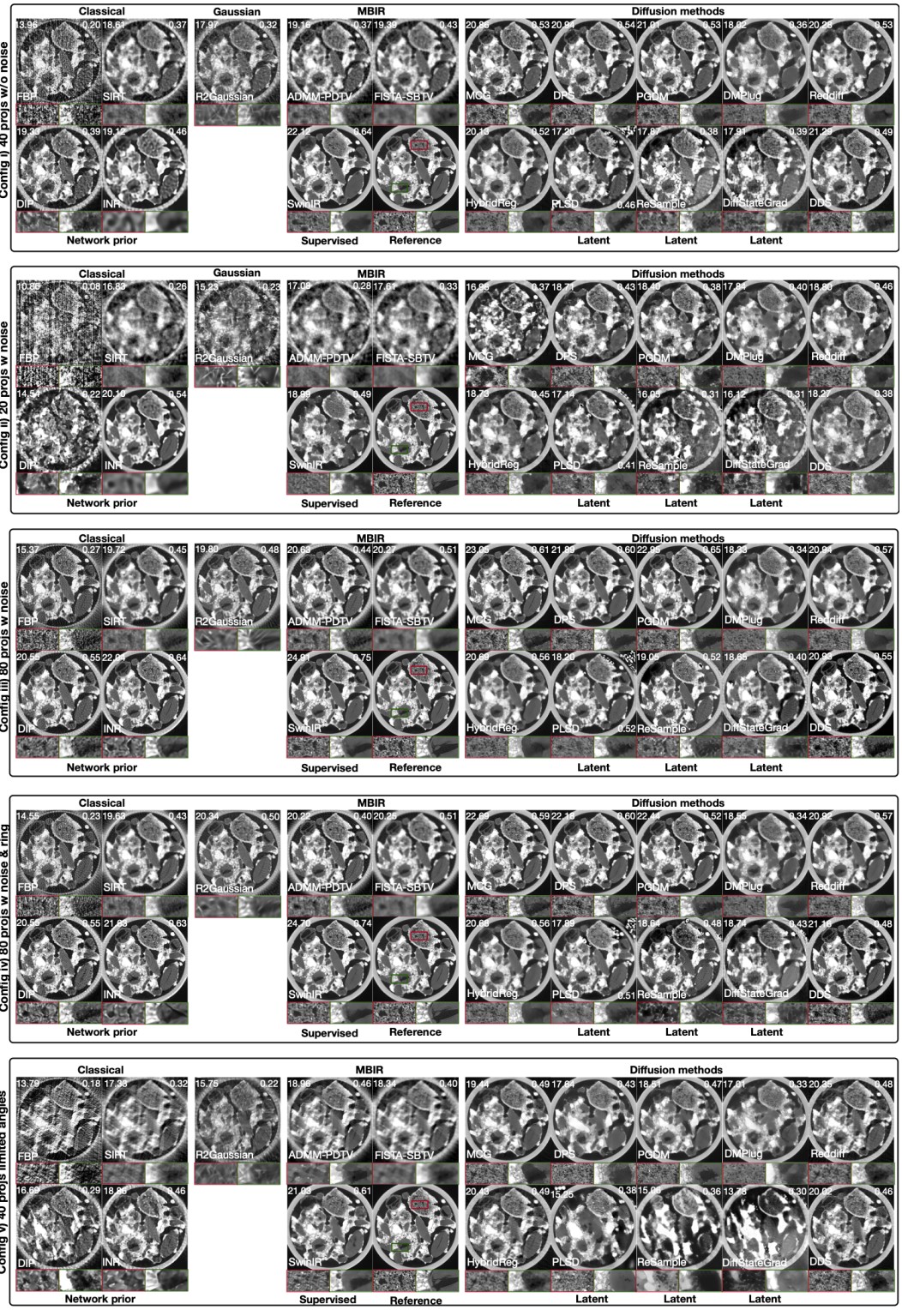

Figure 18: Visual comparison of all benchmarked methods on the simulated industrial CT dataset under four different configurations: (i) 40 projections without noise, (ii) 20 projections with mild noise, (iii) 80 projections with stronger noise, and (iv) 80 projections with noise and ring artifacts. PSNR and SSIM are shown in the top-left and top-right corners. Red and green insets highlight zoomed-in regions.

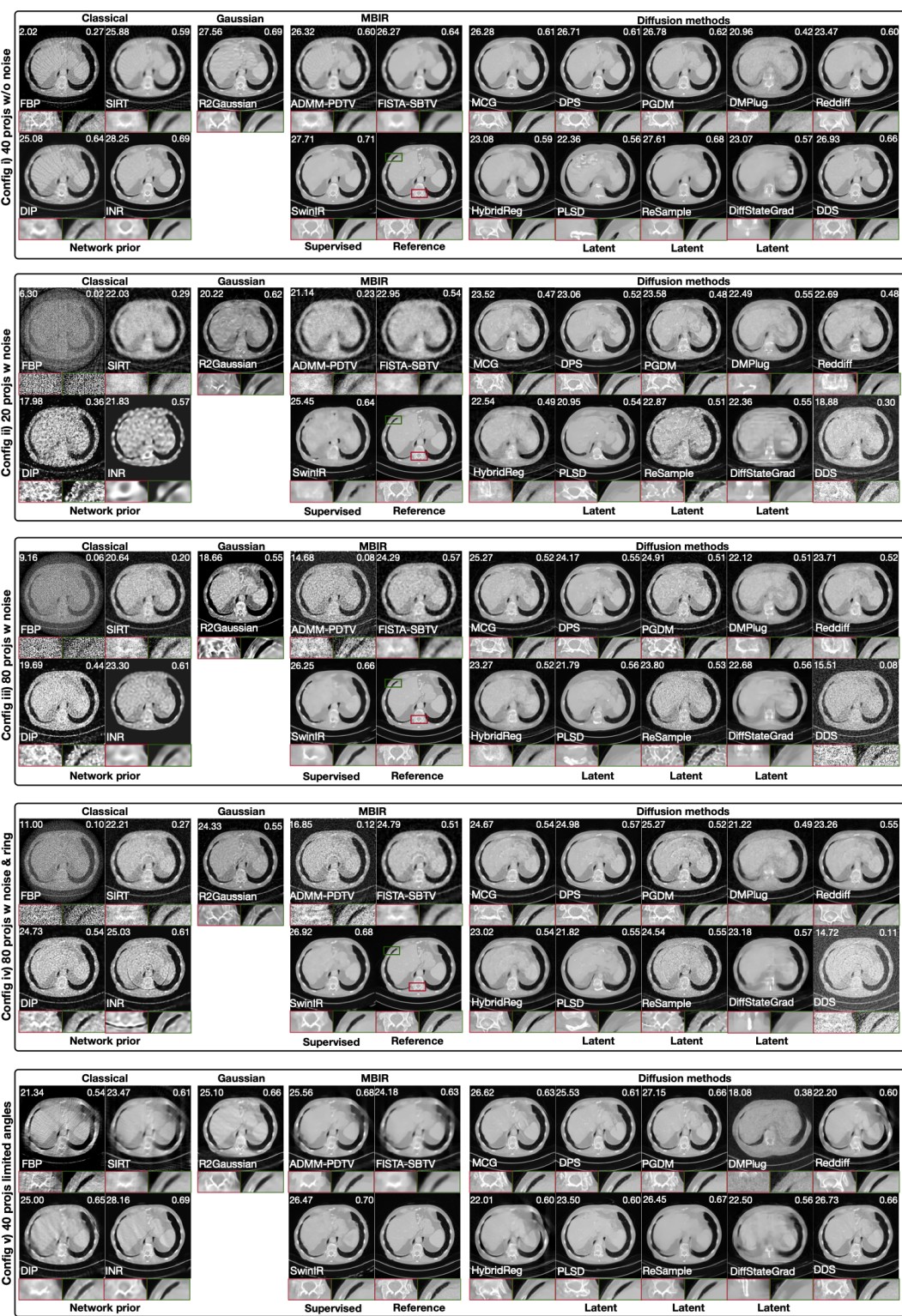

Figure 19: Visual comparison of all benchmarked methods on the simulated medical CT dataset under four different configurations: (i) 40 projections without noise, (ii) 20 projections with mild noise, (iii) 80 projections with stronger noise, and (iv) 80 projections with noise and ring artifacts. PSNR and SSIM are shown in the top-left and top-right corners. Red and green insets highlight zoomed-in regions.

## A.15 IMPLEMENTATION DETAILS

**Diffusion Models.** We implement all diffusion methods using the diffusers library[3]. Training follows (Song et al., 2020; Ho et al., 2020), using a UNet-based architecture as described in (Ronneberger et al., 2015). The detailed UNet configurations used for different datasets are summarized in Table 12, and the VQ-VAE configurations for latent diffusion models are provided in Table 13.

Table 12: UNet configurations used for different CT datasets.

| Parameter | Medical CT | Industrial CT | Synchrotron CT |
|---|---|---|---|
| Input channels | 1 | 1 | 1 |
| Output channels | 1 | 1 | 1 |
| Sample size | 512(pixel)/128(latent) | 512(pixel)/128(latent) | 768(pixel)/192(latent) |
| Activation function | silu | silu | silu |
| Dropout | 0.0 | 0.0 | 0.0 |
| Normalization groups | 32 | 32 | 32 |
| Block output channels | [128, 128, 256, 256, 512, 512] | [128, 128, 256, 256, 512, 512] | [192, 192, 384, 384, 768, 768] |
| Layers per block | 2 | 2 | 2 |
| Down block types | [Down, Down, Down, Down, AttnDown, Down] | [Down, Down, Down, Down, AttnDown, Down] | [Down, Down, Down, Down, AttnDown, Down] |
| Up block types | [Up, AttnUp, Up, Up, Up, Up] | [Up, AttnUp, Up, Up, Up, Up] | [Up, AttnUp, Up, Up, Up, Up] |
| Attention heads | 8 | 8 | 8 |

Table 13: VQ-VAE configurations used in the benchmark for different CT datasets.

| Parameter | Medical CT | Industrial CT | Synchrotron CT |
|---|---|---|---|
| Input channels | 1 | 1 | 1 |
| Output channels | 1 | 1 | 1 |
| Sample size | 512 | 512 | 768 |
| Activation function | silu | silu | silu |
| Block output channels | [128, 256, 512] | [128, 256, 512] | [192, 384, 768] |
| Layers per block | 2 | 2 | 2 |
| Down block types | [DownEnc, DownEnc, DownEnc] | [DownEnc, DownEnc, DownEnc] | [DownEnc, DownEnc, DownEnc] |
| Up block types | [UpDec, UpDec, UpDec] | [UpDec, UpDec, UpDec] | [UpDec, UpDec, UpDec] |
| Mid-block attention | Yes | Yes | Yes |
| Normalization type | group | group | group |
| Normalization groups | 32 | 32 | 32 |
| Latent channels | 1 | 1 | 1 |
| Num VQ embeddings | 512 | 512 | 512 |
| Scaling factor | 1 | 1 | 1 |

**SwinIR.** We implement the SwinIR method using the Hugging Face transformers library[4]. The detailed configurations used for different datasets are summarized in Table 14.

Table 14: SwinIR configurations used in the benchmark for different CT datasets.

| Parameter | Medical CT | Industrial CT | Synchrotron CT |
|---|---|---|---|
| Image size | 512×512 | 512×512 | 768×768 |
| Embed dim | 128 | 128 | 128 |
| Depths | [2, 2, 2, 2] | [2, 2, 2, 2] | [1, 1, 1, 1] |
| Num heads | [2, 2, 2, 2] | [2, 2, 2, 2] | [1, 1, 1, 1] |
| MLP ratio | 2.0 | 2.0 | 1.0 |
| Window size | 8 | 8 | 8 |
| Activation | GELU | GELU | GELU |
| Input channels | 1 | 1 | 1 |
| Output channels | 1 | 1 | 1 |

**INR.** We implement an implicit neural representation (INR) using a SIREN network (Sitzmann et al., 2020), which employs a multilayer perceptron (MLP) with sinusoidal activation functions to model the image as a continuous function over spatial coordinates. The network has a depth of 8 and a hidden layer width of 256. To encode spatial information, we use Fourier feature mapping (Tancik et al., 2020), where a random matrix of shape $\mathbb{R}^{3 \times 256}$ projects the 3D input coordinates to a higher-frequency space. The resulting features are then passed as input to the MLP, allowing the network to represent high-frequency image content effectively. This framework allows direct reconstruction of CT volumes by optimizing the network weights to fit the projection data, without relying on a fixed image grid. Further details on the application of INR to CT reconstruction can be found in (Shen et al., 2022; Wu et al., 2023b;a; Shi et al., 2024b).

**DIP.** For the DIP (Deep Image Prior) method, we adopt a UNet architecture (Ronneberger et al., 2015) as the backbone network. The UNet consists of an encoder with channel sizes 8, 16, 32, and

---

[3] https://github.com/huggingface/diffusers
[4] https://github.com/huggingface/transformers

64, and a symmetric decoder with channel sizes 64, 32, 16, and 8. Skip connections with 4 channels are added at each resolution level. A fixed random noise input, matching the shape of the image to be reconstructed, is fed into the network. The output of the UNet is forward-projected and compared with the actual measurement data. The network parameters are then optimized to minimize this data consistency loss. Despite the absence of external training data, the network's structure alone serves as a strong prior that guides the reconstruction. Further applications of DIP for CT reconstruction can be found in (Baguer et al., 2020; Gong et al., 2018; Barbano et al., 2022; Alkhouri et al., 2024).

**MBIR.** For Model-Based Iterative Reconstruction (MBIR), we use the open-source TomoBAR library[5]. We include two representative total variation (TV)-regularized algorithms: Fast Iterative Shrinkage-Thresholding Algorithm with Primal-Dual TV (FISTA-PDTV) (Beck & Teboulle, 2009; Chambolle & Pock, 2011), and Alternating Direction Method of Multipliers with Split-Bregman TV (ADMM-SBTV) (Boyd et al., 2011; Goldstein & Osher, 2009). These algorithms iteratively minimize a data fidelity term combined with a TV prior to reconstruct the object from projection data.

The regularization weight that balances data fidelity and prior is tuned via grid search using a held-out validation set. All reconstructions are performed using the same geometry and projection operators as in the diffusion-based methods to ensure comparability. We use 200 iterations for both methods and set the regularization iterations at 100.

## A.16 Training Details

**Diffusion Models.** We train both pixel-space and latent-space diffusion models separately for each dataset and use the resulting models as shared backbones across all diffusion-based methods to ensure fair comparisons. Pixel diffusion models are trained for 200 epochs using a batch size of 1 and the AdamW optimizer (Loshchilov & Hutter), with an initial learning rate of $1 \times 10^{-4}$.

For latent diffusion models, we first train a VQ-VAE for 100 epochs using the same optimizer and batch size. Early stopping is applied: training halts if the validation loss does not improve for 10 consecutive epochs. Once the VQ-VAE is trained, we train the UNet in the latent space for 200 epochs using AdamW with the same initial learning rate.

For the real-world dataset, which contains fewer training samples than the simulated datasets, we reduce the learning rate to $1 \times 10^{-5}$ for both pixel and latent diffusion models to prevent overfitting. We apply data augmentation during training by randomly flipping images (horizontally and vertically) and performing random crops between 90% and 100% of the original area, followed by resizing to the original size.

**SwinIR.** The supervised SwinIR model is trained for 200 epochs using AdamW with an initial learning rate of $1 \times 10^{-4}$. Early stopping is employed in the same way as for VQ-VAE. Training typically stops between 120 and 180 epochs based on validation performance.

**INR and DIP.** For INR and DIP, we optimize the network to fit the measured projections for 10,000 iterations using the Adam optimizer (Kingma & Ba, 2014). The learning rate is treated as a tunable hyperparameter, selected separately for each dataset and configuration on held-out subset of the training data.

## A.17 Hyperparameter Selection

To ensure a fair comparison across methods in the DM4CT benchmark, we determine all method-specific hyperparameters through grid search. For each method, we define a search range based on commonly used values in prior work and empirical performance. Ten images are randomly selected from the training dataset of each domain for hyperparameter tuning. Reconstructions are evaluated against corresponding reference images using the mean squared error (MSE), and the hyperparameters yielding the lowest average MSE across the selected images are chosen. The search ranges and selected values for each method are summarized in Table 15.

---

[5]https://github.com/dkazanc/ToMoBAR

Table 15: Search ranges and selected hyperparameters for each reconstruction method in the DM4CT benchmark. Parameters are optimized via grid search on ten randomly selected training images by minimizing average mean squared error with respect to reference reconstructions.

| Method | Parameters | Grid Search Range | Medical CT | Industrial CT | Synchrotron CT |
|---|---|---|---|---|---|
| MCG | step size $\eta$ | $[1\times10^{-4}, 1\times10^3]$ | 0.01 | 0.1 | 0.01 |
| DPS | step size $\eta$ | $[1\times10^{-4}, 1\times10^3]$ | 10 | 1 | 0.05 |
| PSLD | factor on latent error $\gamma$
factor on measurement consistency error $\omega$ | -
 | 0.2/0.2/0.2/0.1
$1-\gamma$ | 0.5/0.5/0.5/0.6
$1-\gamma$ | 0.993
$1-\gamma$ |
| PGDM | step size $\eta$ | $[1\times10^{-4}, 1\times10^3]$ | 1/0.01/0.01/0.01 | 1/0.01/0.01/0.01 | 0.3/0.1/0/1 |
| ReSample | pixel optimization learning rate
latent optimization learning rate | $[1\times10^{-5},1]$
$[1\times10^{-5},1]$ | $1\times10^{-4}/1\times10^{-4}/1\times10^{-3}/1\times10^{-2}$
$0.01/1\times10^{-3}/1\times10^{-3}/1\times10^{-3}$ | $1\times10^{-4}$
0.1/0.01/0.01/0.01 | -
 |
| DMPlug | DDIM steps
learning rate | $[2,3]$
$[1\times10^{-4},1]$ | 3
0.01 | 3
0.01 | 3
0.01 |
| Reddiff | learning rate
factor on measurement consistency error
factor on noise fit error | $[1\times10^{-4},1]$
$[1\times10^{-4},1\times10^6]$
$[1\times10^{-4},1\times10^6]$ | 0.01
0.5/1/1/1
$1\times10^4$ | 0.1/0.01/0.01/0.01
10/1/1/1
$1\times10^3/1\times10^4/1\times10^4/1\times10^4$ | 0.1
1
$2\times10^4/1\times10^4/1\times10^4$ |
| HybridRef | learning rate
factor on measurement consistency error
factor on noise fit error
Portion hybrid noise of previous step | $[1\times10^{-4},1]$
$[1\times10^{-4},1\times10^6]$
$[1\times10^{-4},1\times10^6]$
$(0,1)$ | 0.01
1
$1\times10^4$
0.99/0.999/0.999/0.999 | 0.01
1
-
0.999 | -
-
-
- |
| DiffStateGrad | pixel optimization learning rate
latent optimization learning rate
factor on noise fit error
Variance cutoff for rank adaptation | $[1\times10^{-4},1]$
$[1\times10^{-4},1\times10^6]$
$[1\times10^{-4},1\times10^6]$
$(0,1)$ | 0.01
0.5/1/1/1
$1\times10^4$
0.999/0.999/0.99/0.9999 | 0.1/0.01/0.01/0.01
10/1/1/1
$1\times10^3/1\times10^4/1\times10^4/1\times10^4$
0.999 | -


 |
| INR | learning rate | $[1\times10^{-8}, 1\times10^{-3}]$ | $5\times10^{-6}/1\times10^{-6}/1\times10^{-6}/1\times10^{-6}$ | $1\times10^{-5}/5\times10^{-6}/1\times10^{-5}/1\times10^{-5}$ | $1\times10^{-6}$ |
| DIP | learning rate | $[1\times10^{-8}, 1\times10^{-3}]$ | $5\times10^{-5}/1\times10^{-4}/1\times10^{-4}/1\times10^{-4}$ | $5\times10^{-5}/1\times10^{-4}/1\times10^{-4}/1\times10^{-4}$ | $1\times10^{-5}$ |
| ADMM-PDTV | factor regularization | $[1\times10^{-8},1]$ | 0.01 | 0.01 | 0.005 |
| FISTA-SBTV | factor regularization | $[1\times10^{-8},1]$ | $1\times10^{-3}$ | $1\times10^{-3}$ | $1\times10^{-4}$ |

## A.18 Differentiable CT Forward Operator

To enable CT reconstruction with diffusion models, the forward operator $A$ must be differentiable so that it can be used to enforce data consistency by backpropagating gradients with respect to either the noisy variable $x_t$ or the latent variable $z_t$. Several open-source tomographic toolkits support this functionality, including the ASTRA Toolbox (Van Aarle et al., 2016), the Operator Discretization Library (ODL) (Adler et al., 2017), and the TIGRE toolbox (Biguri et al., 2016), as well as many other CT libraries (Kim & Champley, 2023; Jørgensen et al., 2021). Some of these libraries offer native integration with PyTorch[6], allowing automatic gradient propagation through the projection operator.

In cases where direct integration is not available, a differentiable forward operator can be implemented manually by wrapping the underlying projection and backprojection routines in a custom torch.autograd.Function. This enables seamless integration of the forward model within modern deep learning pipelines. In our implementation, we use the ASTRA Toolbox and show how to wrap its 3D projection and backprojection functionality in a PyTorch-compatible operator. A pseudocode-style listing is provided in Listing 1 to illustrate the approach. The same principle applies to other CT toolkits that expose low-level projection routines.

```python
class OperatorFunction(torch.autograd.Function):
    @staticmethod
    def forward(ctx, volume, projector, projection_shape, volume_shape):
        if volume.ndim == 4:
            projection = torch.zeros((volume.shape[0], *projection_shape)
                , device='cuda')
            for i in range(volume.shape[0]):
                astra.experimental.direct_FP3D(projector, vol=volume[i].
                    detach(), proj=projection[i])
        else:
            projection = torch.zeros(projection_shape, device='cuda')
            astra.experimental.direct_FP3D(projector, vol=volume.detach()
                , proj=projection)
        ctx.save_for_backward(volume)
        ctx.projector = projector
        ctx.volume_shape = volume_shape
        return projection

    @staticmethod
    def backward(ctx, grad_output):
```

---

[6] https://pytorch.org

```python
        volume, = ctx.saved_tensors
        projector = ctx.projector
        volume_shape = ctx.volume_shape
        if volume.ndim == 4:
            grad_volume = torch.zeros((volume.shape[0], *volume_shape),
                device='cuda')
            for i in range(volume.shape[0]):
                astra.experimental.direct_BP3D(projector, vol=grad_volume
                    [i], proj=grad_output[i].detach())
        else:
            grad_volume = torch.zeros(volume_shape, device='cuda')
            astra.experimental.direct_BP3D(projector, vol=grad_volume,
                proj=grad_output.detach())
        return grad_volume, None, None, None

class Operator:
    def __init__(self, vol_geom, proj_geom):
        self.projector = astra.create_projector('cuda3d', proj_geom,
            vol_geom)
        self.volume_shape = astra.geom_size(vol_geom)
        self.projection_shape = astra.geom_size(proj_geom)

    def __call__(self, volume):
        return OperatorFunction.apply(volume, self.projector, self.
            projection_shape, self.volume_shape)

    def T(self, projection):
        volume = torch.zeros(self.volume_shape, device='cuda')
        astra.experimental.direct_BP3D(self.projector, vol=volume, proj=
            projection.detach())
        return volume
```

Listing 1: PyTorch-compatible differentiable ASTRA operator (core logic)

