# OpenReview forum: "DM4CT: Benchmarking Diffusion Models for Computed Tomography Reconstruction"
_ICLR.cc/2026/Conference — ICLR 2026 Poster_

### Official Review · Reviewer_KS57 · 2025-10-31

**Soundness:** 3
**Presentation:** 3
**Contribution:** 4
**Rating:** 6
**Confidence:** 5

**Summary:**

The manuscript introduces **DM4CT**, a comprehensive benchmark for evaluating diffusion-model–based methods for CT reconstruction across medical, industrial, and synchrotron datasets. The work fills an important gap in benchmarking diffusion priors for CT, offering a unified taxonomy, shared backbones, and multi-regime evaluations, including a newly released real synchrotron dataset. The results are insightful: no single diffusion approach dominates, and the trade-off between data fidelity and prior strength is clearly demonstrated. The taxonomy of conditioning strategies—data-consistency gradients, plug-and-play, pseudoinverse guidance, and variational Bayes—is well-structured, and nine diffusion-based and seven classical baselines are compared under a fair setup. Shared latent and pixel diffusion backbones are trained per dataset to ensure fairness, and evaluations span multiple sparsity and noise regimes. Overall, the findings indicate that diffusion models are competitive with classical or MBIR methods but often fall slightly behind a supervised SwinIR baseline in PSNR/SSIM.

**Strengths:**

- **Comprehensive and impressive baselines.**
The range of baselines included is remarkably complete, even surprisingly so. The authors compare not only traditional iterative methods but also a large set of mainstream DM-based inverse solvers, including those not originally designed for CT reconstruction. They further include state-of-the-art supervised models and even classical self-supervised methods such as DIP and INR. This level of comprehensiveness greatly enhances the paper’s value and allows readers to truly understand how much diffusion-based solvers advance CT reconstruction over prior paradigms. Such a thorough comparison is rarely seen in previous work.

- **Unified framework for DM-based inverse solvers.**
A major contribution of this work is the unification and implementation of recent state-of-the-art DM-based inverse solvers under a common interface. Many of these methods have never been applied to CT reconstruction but were validated in other imaging domains. Bringing them together under a consistent framework is a significant service to the CT community and will facilitate the development of more advanced inverse solvers in the future.

**Weaknesses:**

1. **Small-scale training data.**
Although the authors trained both latent and pixel diffusion models within a unified Diffuser framework, a major concern is the extremely small size of the training datasets. Even the largest dataset, AAPM 2016, contains only about 5,000 slices. In contrast, diffusion models used in natural image inverse problems are typically trained on datasets like ImageNet, which contain millions of images. Models trained on such small data are likely to overfit or memorize training samples. While the benchmark may still provide relative performance comparisons among diffusion-based solvers, it cannot accurately reflect their true potential on large-scale data, nor the absolute advantage of diffusion priors over classical or learned reconstruction methods.

2. **Missing key baseline DDS [1].**
Although many DM-based solvers are included, the benchmark omits the crucial **DDS** method. While DDS can loosely be categorized as a pixel-domain data-consistency optimization, it does not strictly follow the plug-and-play paradigm. In fact, DDS empirically finds a strong balance between prior and data consistency through a conjugate-gradient scheme and empirically chosen step size. I strongly recommend including DDS in future comparisons.

3. **Limited diversity of experimental settings.**
Experiments are conducted under only four configurations. Although these capture different regimes, the experimental coverage still feels limited for such an ambitious benchmark.

4. **Limited real-data evaluation.**
The real-projection evaluation uses only two rock samples. I appreciate that the authors conducted real-data experiments, which indeed demonstrate the method’s applicability to high-resolution CT. However, for broader impact, it would be valuable to include real medical CT data. I noticed that the authors conducted medical CT reconstruction experiments in the appendix. However, these experiments mainly serve to illustrate the potential value-range mismatch issue between HU-normalized training and real-world data. Moreover, only quantitative results are provided, without any qualitative visualizations. I encourage the authors to include more comprehensive experiments on real medical projection data.

5. **Limited evaluation metrics.**
The study mainly reports PSNR and SSIM. Although these are standard metrics, they do not capture diagnostic quality, which is critical in medical CT. For example, in Figure 13, particularly under Config 2 and 3, the supervised method achieves the highest PSNR and SSIM but produces reconstructions that are severely blurred and anatomically meaningless—an instance of “high scores but clinically useless results.” While the appendix includes LPIPS and data-fit metrics, these still fail to reflect whether key anatomical structures are correctly reconstructed. I encourage the authors to include more diverse evaluation strategies, such as downstream segmentation accuracy, organ-volume error, or even radiologist scoring.

6. **Limited generalization analysis.**
The paper does not address an essential aspect of generalization—cross-dataset robustness. I encourage the authors to evaluate across datasets collected under different protocols or vendor settings. Since AAPM 2016 volumes are relatively homogeneous, this would significantly enhance the completeness of the benchmark.

> [1] Chung, Hyungjin, Suhyeon Lee, and Jong Chul Ye. *“Decomposed Diffusion Sampler for Accelerating Large-Scale Inverse Problems.”* ICLR, 2024.

---
Despite these issues, the paper remains a valuable and well-structured contribution. I encourage the authors to address these concerns in a revised version, which would substantially improve the paper’s quality and impact. If these improvements are made, I would consider increasing my score. I genuinely hope this line of inquiry will continue, as it has strong potential to advance CT reconstruction research.

**Questions:**

1. For the medical CT configuration (Config 1), none of the methods seem to produce reasonable results for 40-view noiseless sparse-view reconstruction. Surprisingly, most DM-based methods almost completely fail, while the self-supervised INR baseline—though blurry—still reconstructs the main anatomical structures. Is such failure expected in this simplest setup?
2. The INR-based approach consistently performs well across CT scenarios, especially in low-noise and real-data regimes. In Figure 2 (industrial Config 2), it is the only method that correctly reconstructs the main structural shape, while all others fail. This suggests that INR has a unique advantage in preserving structure, albeit at the cost of sharpness. This naturally raises the question: could combining INR with diffusion priors yield further gains? Recent works, such as **DPER [2]**, have explored this direction, and it may be valuable for the authors to discuss or reflect on this possibility.
3. In my own experience training diffusion models on the AAPM 2016 dataset, I observed an interesting phenomenon: with such a small dataset (~ 5k slices), diffusion models overfit easily. Early-epoch models generate poor unconditional samples but produce better reconstructions when used as priors, while later-epoch models generate visually impressive samples yet degrade in reconstruction quality. This effect lessens on larger datasets (~100k slices). I encourage the authors to verify whether this occurs in their experiments, as it could be a useful observation and reinforces the need for larger-scale training.
4. There is a typo in line 420 of the manuscript: “Figure 6a” and “Figure 6b” should be corrected to “Figure 7a” and “Figure 7b,” respectively.

> [2] Du, Chenhe, et al. *“DPER: Diffusion Prior Driven Neural Representation for Limited-Angle and Sparse-View CT Reconstruction.”* arXiv:2404.17890 (2024).

---

> ### Author Response · Authors · 2025-11-24
>
> We thank the reviewer for the constructive comments and address each point in turn.
>
> > Small-scale training data.
>
> We fully agree that the available training data are small compared to the large-scale natural-image datasets commonly used to train diffusion models (e.g., ImageNet-scale). Unfortunately, this limitation reflects the reality of CT imaging rather than a design choice.
>
> In medical CT, patient-privacy restrictions and institutional approvals severely constrain the release of large public datasets. In industrial CT, the lack of standardized calibration, scanning conditions, and proprietary acquisition pipelines make cross-machine aggregation difficult. As a result, diffusion models (and all comparison methods) must operate under inherently data-limited conditions. Our benchmark is therefore designed to reflect this practical constraint rather than an idealized scenario.
>
> To partially mitigate this issue, we contribute a high-resolution synchrotron dataset, acquired under strictly controlled and repeated conditions using similar objects. This provides a cleaner, more consistent training environment compared to typical industrial CT, and represents a step toward more reliable benchmarking.
>
> We also conducted additional experiments on fine-tuning pretrained natural-image autoencoders (from SDXL) on CT images in alliance with inquery from reviewer MUjs. While this shows some promise, our findings indicate that adapting such models to low-diversity, domain-specific CT data is nontrivial and remains an interesting direction for future research. Full details are provided in Appendix A.11.
>
> > Missing key baseline DDS.
>
> We have now included DDS as a full baseline in our benchmark (Figure 2 and Table 2 in the main paper; Tables 10–11 and Figures 17–19 in the appendix). Across our experiments we find that DDS performs well under noiseless scenarios but degrades significantly under Poisson noise, which is the dominant noise model for CT. We discuss this behavior in detail in Appendix A.9.
>
> Conceptually, DDS reconstructs noisy measurements via the linear system $(\gamma A^{\mathsf T} R A + I)x = \hat{x}_0 + \gamma A^{\mathsf T} R y$, which corresponds to a Gaussian likelihood with inverse covariance  $R = \Sigma^{-1}$.
>
> * For additive Gaussian noise with variance $\sigma^2$, $R=\frac{1}{\sigma^2}I$. So DDS reduces to tuning a single scalar hyperparameter $\sigma$. This is the regime DDS is designed for, and we indeed observe strong performance.
> * For Poisson noise, however,
> $\mathrm{Var}(y_i) \approx I_0 e^{-(Ax)_i}$, which means $R = \mathrm{diag}\left(\frac{1}{\sigma_i^2}\right)$. It becomes data-dependent, spatially varying, and intensity-dependent. It cannot be reduced to a single scalar hyperparameter. This mismatch with the underlying noise model explains DDS’s degraded performance in the CT setting.
>
> To illustrate this clearly, we compare Gaussian and Poisson noise (Table below). We simulate similar noise levels such that they achieve similar FBP reconstruction PSNR/SSIM. DDS performs well for Gaussian noise but drops sharply for Poisson noise. Figure 11 (Appendix) provides visual comparisons showing that DDS reconstructs Gaussian-corrupted measurements well but struggles with Poisson-corrupted ones.
>
> |  | FBP         | DDS         |
> |--------------|-------------|-------------|
> | No extra noise   | 26.45/0.68  | 30.08/0.83  |
> | Gaussian noise  | 22.11/0.41  | 28.54/0.76  |
> | Poisson noise  | 22.27/0.42 | 25.00/0.64  |

---

> ### Author Response · Authors · 2025-11-24
>
> > Limited diversity of experimental settings.
>
> Our goal was to benchmark diffusion-based CT reconstruction under representative and commonly studied regimes in the CT literature. The four configurations in the initial submission—sparse-view, ultra–sparse-view, noisy, and noise + artifacts—were chosen because they correspond to the most practically relevant challenges in both medical and industrial CT.
>
> Following the reviewer’s suggestion, we have now added an additional experimental setting using 40 limited-angle projections over the range
> $(0,3/4\Pi]$. These results are included in the revised manuscript (Table 2, Tables 10–11, Figures 18–19). Under this configuration, diffusion methods outperform supervised SwinIR but still underperform INR on the medical dataset.
>
> We also note that existing CT benchmark studies typically evaluate fewer settings or datasets than those considered in DM4CT (e.g., [1–3]). Relative to prior benchmarks, DM4CT covers more datasets, more acquisition regimes, and a wider variety of modern reconstruction algorithms.
>
> Exsiting benchmark papers on CT often are conducted with fewer experimental settings or datasets than we do [1,2,3]. We believe our current experimental settings and datasets are sufficient to provide insights in the paper. Overall, we believe the experimental settings in DM4CT are sufficiently diverse to provide meaningful insights,
>
> [1] Eulig et al., Benchmarking deep learning-based low-dose CT image denoising algorithms, Medical Physics 2024
> [2] Shan et al., Competitive performance of a modularized deep neural network compared to commercial algorithms for low-dose CT image reconstruction, Nature Machine Intelligence 2019
> [3] Kiss et al., Benchmarking learned algorithms for computed tomography image reconstruction tasks, Applied Mathematics for Modern Challenges 2025
>
> > Limited real-data evaluation.
>
> We appreciate the reviewer’s concern. The use of real synchrotron data in DM4CT is intentional and motivated by practical constraints in medical CT reconstruction research.
>
> First, real medical projection datasets with complete acquisition geometry are extremely scarce. Most publicly available medical CT datasets only provide reconstructed volumes, not raw projections or system parameters. When projection data exist, they are typically acquired with helical cone-beam geometries, which introduce two major obstacles for diffusion-model benchmarking:
>
> * Full 3D reconstruction is required, making both training and inference prohibitively expensive for diffusion models.
> * Training data becomes extremely limited. For instance, the AAPM dataset contains in total 20 patient volumes, which is inadequate for training diffusion models. In contrast, using 2D slices yields tens of thousands of training samples.
>
> We performed additional experiments in the appendix using the AAPM 2016 dataset by rebinning helical projections into cone-beam geometry.
> We now added the missing visualization for these experiments in Figure 10. However, these experiments mainly serve to illustrate value-range mismatch issues between HU-normalized training and real-world measurements. Moreover, rebinning introduces missing-wedge artifacts near the volume boundaries, leaving only a small central region usable for benchmarking. This further reduces the effective number of training and test slices.
>
> To overcome these limitations, we acquired and released a new high-resolution synchrotron CT dataset in our DM4CT benchmark included in the initial submission. This dataset uses a simple parallel-beam, circular trajectory, offering several advantages:
> * Accurate slice-wise 2D reconstruction, enabling systematic evaluation across many slices;
> * Greatly reduced computational cost, which is essential for evaluating diffusion-model pipelines;
> * High photon flux and fine spatial resolution, providing a realistic but controlled environment to stress-test reconstruction algorithms.
>
> For these reasons, synchrotron data offer a practical and scientifically valuable platform for benchmarking diffusion methods, especially given the computational and data limitations inherent in real medical CT acquisition.

---

> ### Author Response · Authors · 2025-11-24
>
> > Limited evaluation metrics.
>
> In DM4CT, our goal is to benchmark reconstruction methods rather than to assess clinical diagnostic quality. For this reason, we primarily report PSNR and SSIM, which remain standard accuracy metrics in the CT reconstruction literature. Clinical metrics, such as radiologist scoring or pathology detection, require dedicated study design, expert involvement, and often domain-specific imaging protocols, which are outside the scope of this benchmark.
>
> That said, we fully agree that high PSNR/SSIM does not necessarily imply clinically meaningful reconstruction. Following the reviewer’s suggestion, we conducted an initial downstream-task evaluation using segmentation accuracy. Specifically, we used the Segment Anything Model (SAM) to segment both the reference slices and the reconstructed slices, and treated the SAM segmentation of the reference slices as “ground truth” for quantifying segmentation quality. Full results are included in Appendix A.10 (Table 6, Figure 12).
>
> Our findings show that better reconstruction metrics do not guarantee better downstream segmentation. Some diffusion-based reconstructions achieve high PSNR/SSIM but yield weaker overlap with SAM-generated reference masks, while certain classical or INR-based methods segment more reliably despite lower pixel-wise metrics.
>
> However, we emphasize that these segmentation experiments serve only as a preliminary investigation. The results depend strongly on, for example:
>
> * the choice of segmentation model (SAM was trained on natural images, not CT),
> * the mask-matching protocol needed to compare instances across methods.
>
> Therefore, we caution against drawing clinical conclusions from these results. A more rigorous clinical downstream evaluation (e.g., organ-level metrics, volumetric error, or radiologist studies) is an important direction for future work. We include the discussion into the limitation and future work section.
>
>
> | Metric (Dice / IoU) | FBP | SIRT | ADMM-PDTV | FISTA-SBTV | DIP | INR | R2 Gaussian | SwinIR | MCG | DPS | PSLD | PGDM | Resample | DMPlug | Reddiff | DiffStateGrad | DDS | HybridReg |
> |---------------------|-----|-------|-----------|-------------|------|------|-------------|---------|------|------|--------|--------|-----------|---------|-----------|----------------|-------|-----------|
> | **Dice / IoU**      | 0.603 / 0.550 | 0.819 / 0.735 | 0.781 / 0.707 | 0.418 / 0.359 | 0.345 / 0.283 | 0.707 / 0.633 | 0.551 / 0.516 | 0.683 / 0.636 | 0.604 / 0.533 | 0.649 / 0.585 | 0.381 / 0.320 | 0.635 / 0.565 | 0.623 / 0.548 | 0.497 / 0.403 | 0.493 / 0.414 | 0.385 / 0.301 | 0.717 / 0.657 | 0.485 / 0.405 |
>
> > Limited generalization analysis.
>
> We agree with the reviewer that cross-dataset generalization is an important and valuable direction. Our current benchmark focuses primarily on within-domain comparisons (medical, industrial, synchrotron), with the goal of evaluating diffusion-based CT reconstruction methods under consistent acquisition conditions. As a result, we did not perform extensive cross-dataset robustness experiments in this version of the benchmark. We have now added this discussion to the “Future Work” section (line 501-512) and plan to explore cross-dataset generalization in the follow-up research.
>
> >Q: For the medical CT configuration (Config 1), none of the methods seem to produce reasonable results for 40-view noiseless sparse-view reconstruction. Surprisingly, most DM-based methods almost completely fail, while the self-supervised INR baseline—though blurry—still reconstructs the main anatomical structures. Is such failure expected in this simplest setup?
>
> Yes. This behavior is expected, and it reflects a fundamental difference between INR and prior-driven diffusion reconstruction in the noiseless, extremely sparse-view regime.
>
> In the 40-view noiseless setting, the inverse problem is still severely ill-posed. There exist many images that perfectly match the sinogram, and the measurements alone do not uniquely determine the ground truth.
>
> INR fits the measurements directly by optimizing only the data-consistency term. Since it is not constrained by a learned prior, it converges to a solution that matches the projections—often blurry, but structurally correct.
>
> In contrast, diffusion-based methods solve a different optimization objective: they reconstruct an image that simultaneously (1) fits the measurements and (2) remains close to the learned diffusion prior. The strength of that prior–consistency tradeoff is method-dependent (and hyperparameter-dependent). In extremely sparse and noiseless scenarios, diffusion solvers tend to select a reconstruction within the manifold learned from training data and simultaneously consistent with the measurements. This may deviate in fine details from the reference image even though it is consistent with the measurements.

---

> ### Author Response · Authors · 2025-11-24
>
> > Q: The INR-based approach consistently performs well across CT scenarios, especially in low-noise and real-data regimes. In Figure 2 (industrial Config 2), it is the only method that correctly reconstructs the main structural shape, while all others fail. This suggests that INR has a unique advantage in preserving structure, albeit at the cost of sharpness. This naturally raises the question: could combining INR with diffusion priors yield further gains? Recent works, such as DPER [2], have explored this direction, and it may be valuable for the authors to discuss or reflect on this possibility.
>
> We fully agree that INR demonstrates strong structural preservation, particularly in low-noise and real-data regimes, and that combining INR with diffusion priors is a highly promising research direction.
>
> Conceptually, INR performs well in reconstruction because it enforces strong spatial smoothness and continuity through its implicit representation, which naturally preserves global structure even under severe sparsity. In contrast, diffusion priors offer powerful data-driven regularization, capturing fine-scale textures and statistical characteristics learned from the training set. A hybrid INR–diffusion approach could in principle leverage the strengths of both: INR’s structural correctness and diffusion’s expressive learned prior.
>
> However, we notice that NR-based optimization is computationally expensive. Practical integration with diffusion models would therefore require careful design to ensure efficiency. Recent work such as DPER indeed supports the potential of this direction, and we now include a discussion of such INR–diffusion hybrids in the revision.
>
>
> > Q: In my own experience training diffusion models on the AAPM 2016 dataset, I observed an interesting phenomenon: with such a small dataset (~ 5k slices), diffusion models overfit easily. Early-epoch models generate poor unconditional samples but produce better reconstructions when used as priors, while later-epoch models generate visually impressive samples yet degrade in reconstruction quality. This effect lessens on larger datasets (~100k slices). I encourage the authors to verify whether this occurs in their experiments, as it could be a useful observation and reinforces the need for larger-scale training.
>
> We also found it extremely interesting and conducted an additional experiment to verify it. The results have been added in Appendix A.13 (Table 9 and Figure 15).
>
> | Stage | Early            | Mid        | Final            |
> |-------|------------------|------------|------------------|
> | PSNR / SSIM | **30.68 / 0.75** | 28.46 / 0.72 | 28.71 / 0.73 |
>
> We confirm with the reviewer that we observe the same behavior on the AAPM dataset. In our experiment, we train a pixel-space diffusion model for 200 epochs and evaluate checkpoints at 25 epochs (early stage), 100 epochs (mid stage), and 200 epochs (final stage). Each checkpoint is used both for unconditional generation and for CT reconstruction.
>
> Our findings:
>
> * Early-stage diffusion models fail in unconditional generation, producing noisy samples.
> * Mid-stage and final-stage models both generate visually plausible unconditional samples.
> * Despite the lack of unconditional generation capability, the early-stage model achieves the best CT reconstruction, both quantitatively and visually.
> * Mid- and final-stage models reconstruct similar CT images, with the final-stage model yielding slightly higher PSNR/SSIM but still below the early-stage model by large margin.
>
> This is a highly intriguing phenomenon and aligns with the reviewer’s prior experiences. It suggests that, in small-data regimes stronger fitting to the training distribution may hurt inverse-problem performance, whereas early-stage models retain a more flexible and less overfitted prior. In this case, early stopping reduces training cost by 87.5%, but further empirical investigation is required to understand (1) when to early stop, (2) whether it holds across domains, and (3) what principles govern the optimal stopping point. We plan to explore these questions in future work.
>
> > There is a typo in line 420 of the manuscript: “Figure 6a” and “Figure 6b” should be corrected to “Figure 7a” and “Figure 7b,” respectively.
>
> We fixed the typos.

---

### Official Review · Reviewer_L1Ta · 2025-11-01

**Soundness:** 2
**Presentation:** 3
**Contribution:** 2
**Rating:** 2
**Confidence:** 4

**Summary:**

The paper gives a summary of nine diffusion models for CT reconstruction. Three datasets are included in the paper. However, there is not any new method or improvement was proposed and only one dataset was given.

**Strengths:**

The paper presents a comprehensive and well-organized summary and comparison of nine existing algorithms within the domain.

1 All the nine methods were evaluated and extensive experiments were performed. The paper gives comparison with other method and the quantitive result.

2 Comparison and comprehension of nine diffusion-based methods were given. Every dataset was evaluated with different configuration.

3 A high-resolution synchrotron dataset was provided. The dataset is a new dataset for reconstruction.
.

**Weaknesses:**

1 Lack of Novelty and Original Contribution: The paper primarily focuses on summarizing, comparing, and reproducing results of existing algorithms. The paper gives the result and summary of different papers. The main work is the running and comparison of different method. There is not any new design of the model.

2 Limited new understanding for future research of diffusion model in CT reconstruction. Better suited for publication as a survey paper or technical report.

3. More details of the high-resolution synchrotron dataset is needed.

**Questions:**

Inference time very important for diffusion model needs sampling. Add some analysis of inference is preferred.

---

> ### Author Response · Authors · 2025-11-24
>
> > Lack of Novelty and Original Contribution: The paper primarily focuses on summarizing, comparing, and reproducing results of existing algorithms. The paper gives the result and summary of different papers. The main work is the running and comparison of different method. There is not any new design of the model.
>
> We would like to emphasize that **our goal is not to propose a new reconstruction algorithm, but to provide the first systematic benchmark for diffusion models in CT**. This is explicitly stated in our introduction  (the last paragraph and stressed by making it italic). Our contributions are original within the scope of a benchmark paper and go well beyond simply “running and comparing” existing methods.
>
> First, we introduce DM4CT, the first comprehensive benchmark that evaluates a broad family of diffusion-based CT reconstruction approaches under controlled simulation settings and real experimental conditions.
>
> Second, we acquire and release a new high-energy synchrotron CT dataset, which provides a unique, high-resolution testbed well suited for evaluating diffusion-based reconstruction methods in practice.
>
> Third, we propose a unified taxonomy that organizes diffusion methods by their data-consistency strategies and use of priors, clarifying conceptual relationships between methods that were previously fragmented across papers.
>
> Fourth, we implement all methods in the widely used diffusers framework and open-source a clean, unified codebase
> [https://github.com/DM4CT/DM4CT](https://github.com/DM4CT/DM4CT).
> As Reviewer MUjs noted:
> ```
> "The codebase is maintained very well and easy to use. The code is easy to follow and I can switch between different methods. The code style is great and different classes of methods are separated clearly. Furthermore, it combines a lot of algorithms very simply in just a couple of python files."
> ```
>
> We perform extensive experiments providing practical insights into the strengths, limitations, and deployment challenges of diffusion models in CT. We provide various insights in different aspects, such as tradeoff between prior and consistency, discussion on data-consistency approaches specifically for latent diffusion models, computation efficiency, etc.
>
> Based on the discussion with other reviewers, we also include additional experiments for several interesting phenomena observed during rebuttal. To shortly summarize:
>
> * We perform segmentation as a downstream task, and show that high-quality diffusion-based reconstructions do not necessarily guarantee strong segmentation performance.
> * We find that diffusion models trained at early stages can yield better CT reconstruction, despite lacking unconditional generation capability.
> * We further explore the balance between pixel and latent optimization under different noise scenarios.
> * We fine-tune existing pretrained autoencoders and find that adapting natural-image encoders to CT is not a trivial task.
>
> All these above are our original contributions.

---

> ### Author Response · Authors · 2025-11-24
>
> > Limited new understanding for future research of diffusion model in CT reconstruction. Better suited for publication as a survey paper or technical report.
>
> We respectfully disagree with the characterization of this work as a survey. Our contribution is not a summary of existing literature, but a new, experimentally grounded benchmark framework that produces insights not available in prior work.
>
> ICLR explicitly welcomes papers in the Datasets and Benchmarks track, and our submission fits this category: it introduces a new real-world dataset, unifies and re-implements disparate diffusion-based CT reconstruction methods in a standardized framework, and provides systematic experimental comparisons that reveal behaviors, failure modes, and trade-offs that have not been reported before.
>
> Our benchmark yields new technical findings directly relevant to future research, including:
>
> * practical trade-offs between data-consistency strategies in latent vs. pixel space,
> * the interaction between noise levels and optimization dynamics,
> * early-stage diffusion models outperforming fully trained ones for CT reconstruction,
> * limitations of existing pretrained natural-image encoders when adapted to CT,
> * and downstream-task discrepancies (e.g., segmentation robustness).
>
> We therefore believe ICLR is the correct venue for this paper and the chosen "datasets and benchmarks" is the correct area. The recent ICLR literature includes many accepted benchmark papers demonstrating that rigorous, carefully designed benchmarks can make substantial scientific contributions by shaping future research directions. Our submission aims to play this role specifically for diffusion models in CT.
>
> Examples of benchmark papers published in ICLR:
>
> Wu et al., Q-Bench: A Benchmark for General-Purpose Foundation Models on Low-level Vision, ICLR 2024
> Zhang et al., MIntRec2.0: A Large-scale Benchmark Dataset for Multimodal Intent Recognition and Out-of-scope Detection in Conversations, ICLR 2024
> Shern et al., MLE-bench: Evaluating Machine Learning Agents on Machine Learning Engineering, ICLR 2025
> Zheng et al., InverseBench: Benchmarking Plug-and-Play Diffusion Priors for Inverse Problems in Physical Sciences, ICLR 2025
> ...
>
> > More details of the high-resolution synchrotron dataset is needed.
>
> We appreciate the reviewer’s interest in the synchrotron dataset. In the initial submission, we tried to provide all scientifically relevant details required to understand and reproduce the data, including the full set of acquisition parameters (e.g., beam energy, detector specifications, number of projections, exposure time) and all reconstruction-critical parameters (such as off-rotation center), along with a script that can directly reconstruct the dataset from raw projections. These materials are included in both the paper and the accompanying codebase: [https://github.com/DM4CT/DM4CT](https://github.com/DM4CT/DM4CT).
>
> We agree that more information is better, but some information, such as the specific synchrotron facility at which the scan was performed, was intentionally omitted during the review period. Because there are relatively few such facilities worldwide, disclosing this would compromise anonymity. This information will be fully included upon camera-ready submission.
>
> We hope this clarifies that the dataset description is complete for reproducibility while preserving anonymity during review.
>
> > Inference time very important for diffusion model needs sampling. Add some analysis of inference is preferred.
>
> We agree that inference efficiency is an important aspect of diffusion-based reconstruction. In fact, our original submission already includes a dedicated Computation Efficiency section, where we report and compare:
> * the sampling time of every diffusion method,
> * memory consumption,
> * as well as the training cost of each model-based and latent-diffusion approach.
>
> These results quantify the computational trade-offs across methods and help establish practical guidance for future research.
>
> **We kindly ask the reviewer to consider our clarifications and additional experiments when reassessing the manuscript, and we appreciate any further suggestions that could help improve the work.**

---

### Official Review · Reviewer_MUjs · 2025-11-01

**Soundness:** 4
**Presentation:** 4
**Contribution:** 3
**Rating:** 6
**Confidence:** 4

**Summary:**

This paper provides a benchmark on CT reconstruction algorithms on different datasets and settings.

**Strengths:**

1. As far of my knowledge, this paper provides a first comprehensive benchmark of current SOTA CT reconstruction methods. This benchmark is very comprehensive, consisting of INR-based methods, pixel-diffusion methods, latent-diffusion methods, MBIR methods, traditional methods, and transformer-based methods.
2. The codebase is maintained very well and easy to use. The code is easy to follow and I can switch between different methods. The code style is great and different classes of methods are separated clearly. Furthermore, it combines a lot of algorithms very simply in just a couple of python files.
3. The benchmark on different datasets is comprehensive and easy to interpret.
4. The analysis is also insightful. I appreciate the discussion on the tradeoff of latent gradient-based approach and external optimization.

**Weaknesses:**

1. Including some GAN-based methods will strengthen this paper. If authors have time, I would also encourage trying some Gaussian splat based methods as these methods are gaining more popularity recently.
2. If authors have time, I am also interested in the performance of flow-based methods, such as FlowDPS or so on.
3. I would encourage the authors to fine tune from some pretrained autoencoder, specifically that from SDXL or SD3, instead of training from scratch, as training from scratch may lose some generalization capability.
4. For external consistency optimization, there is usually a necessity of using early stopping by noise level adaptively based on diffusion timestep. Also, the choice of latent optimization v.s. pixel optimization can be crucial (more noise level probably more latent optimzation). I understand that this setting can be tricky, but I encourage authors to adopt some hyperparameter tuning to see whether this will solve the artifact problem in the external-consistency relied methods (Especially for latent diffusions).

**Questions:**

I do not have other questions.

---

> ### Author Response · Authors · 2025-11-24
>
> We thank the reviewer for recognizing the contributions of our work and for providing thoughtful and insightful comments. We address each point of the reviewer below.
>
> > I would encourage the authors to fine tune from some pretrained autoencoder, specifically that from SDXL or SD3, instead of training from scratch, as training from scratch may lose some generalization capability.
>
> We conducted a dedicated experiment to evaluate this aspect, now included in Appendix A.11 (Table 7, Figure 13).
> SDXL and SD3 are latent diffusion models trained on large-scale natural-image corpora, and both use a KL-regularized variational autoencoder (AutoencoderKL). In our study, we adopt the SDXL AutoencoderKL released by stabilityai and evaluate three settings:
> 	1.	Directly using the pretrained AutoencoderKL (no fine tuning)
> 	2.	Fine tuning the pretrained AutoencoderKL on the full CT training set
> 	3.	Training the AutoencoderKL from scratch on CT images
>
> Since SDXL’s autoencoder expects RGB input, CT slices are duplicated across three channels. After training each autoencoder, we train a separate latent diffusion model for them.
>
> ### Representation (autoencoder reconstruction) and CT reconstruction quality
>
> | Autoencoder type             | Config           | Training epochs | Autoencoder recon. | CT recon.        |
> |------------------------------|------------------|------------------|---------------------|-------------------|
> | AutoencoderKL (SDXL)         | w/o fine tuning  | 0                | 37.12 / 0.88        | 21.34 / 0.42      |
> | AutoencoderKL (SDXL)         | w fine tuning    | 5                | 37.13 / 0.90    | 23.97 / 0.66  |
> | AutoencoderKL (SDXL)         | from scratch     | 200              | 34.70 / 0.88        | 23.79 / **0.72**|
> | VQ-VAE (this benchmark)      | from scratch     | 200              | **39.30 / 0.92**    | **25.52** / 0.70|
>
> ### Representation (autoencoder reconstruction)
> Our findings:
>
> * Even without fine tuning, the pretrained AutoencoderKL represents CT images reasonably well.
> * Fine tuning yields a small but consistent improvement over direct use.
> * Training AutoencoderKL from scratch gives the lowest representation ability.
> * VQ-VAE used in the benchmark achieves the strongest representation quality (highest PSNR/SSIM).
>
> These differences relate to architectural properties: KL-regularized VAEs enforce a Gaussian latent prior, which helps on large, diverse datasets but can degrade under limited data or low diversity, as often occurs in CT. This is consistent with observations in prior work[1,2].
>
> [1] Xiao et al., A Note on Generalization in Variational Autoencoders: How Effective Is Synthetic Data and Overparameterization?, TMLR 2025
> [2] Wang et al., Posterior collapse and latent variable non-identifiability, NIPS2021
>
> ### Unconditional generation and CT reconstruction
>
> We also evaluate unconditional generation and CT reconstruction using the corresponding latent diffusion models:
>
> * The pretrained AutoencoderKL (no fine tuning) reconstructs well as an autoencoder but fails during unconditional generation and CT reconstruction (producing unstable or noisy results).
> * The fine-tuned AutoencoderKL generates reasonable samples but still somewhat blurry.
> * The AutoencoderKL trained from scratch generates high-quality unconditional samples and provides the best CT reconstruction performance among the AutoencoderKL variants.
> * The VQ-VAE used in our benchmark yields the highest CT reconstruction PSNR/SSIM overall.
>
> Visual comparisons are included in Figure 13.
>
> ### Conclusion
>
> Fine-tuning natural-image autoencoders is a promising direction and does work to some extent.
> However, in our experiments:
>
> * The fine-tuned AutoencoderKL still underperforms compared to a task-specific autoencoder trained from scratch.
> * This outcome likely depends on autoencoder architecture, training scale, and the substantial structural/statistical differences between CT and natural images.
> * More sophisticated domain-adaptation strategies may be needed for optimal performance.
>
> We plan to explore these directions in future work and include this into the future work section.

---

> ### Author Response · Authors · 2025-11-24
>
> > For external consistency optimization, there is usually a necessity of using early stopping by noise level adaptively based on diffusion timestep. Also, the choice of latent optimization v.s. pixel optimization can be crucial (more noise level probably more latent optimzation). I understand that this setting can be tricky, but I encourage authors to adopt some hyperparameter tuning to see whether this will solve the artifact problem in the external-consistency relied methods (Especially for latent diffusions).
>
> We conducted an ablation study on data-consistency optimization, focusing on the interaction between pixel-space optimization, latent-space optimization, and noise level, and we now include these results in Appendix A.12 (Table 8, Figure 14).
> In this study, we fix the learning rates for both pixel and latent optimization and vary only the number of optimization iterations.
>
> #### config iii)
>
> | iters | 25   (pixel)        | 50  (pixel)         | 100  (pixel)            | 200 (pixel)          |
> |--------------|-------------|-------------|------------------|--------------|
> | 100  (latent)   | 27.91/0.78  | 28.79/0.79  | **29.23/0.79**  | 28.90/0.77   |
> | 200   (latent)  | 27.90/0.78  | 28.76/0.79  | 29.20/0.79      | 28.87/0.77   |
>
>
> #### config iii – more noise
> | Latent iters | 25  (pixel)         | 50 (pixel)          | 100 (pixel)             | 200  (pixel)         |
> |--------------|-------------|-------------|------------------|--------------|
> | 100  (latent)    | 27.64/0.77  | 28.40/0.77  | 28.23/0.75      | 27.77/0.72   |
> | 200 (latent)    | 27.75/0.77  | 28.38/0.78  | **28.40/0.78**  | 27.65/0.72   |
>
>
> #### config iii – less noise
> | Latent iters | 25  (pixel)         | 50  (pixel)         | 100  (pixel)            | 200   (pixel)           |
> |--------------|-------------|-------------|------------------|------------------|
> | 100   (latent)    | 27.95/0.78  | 29.06/0.80  | 29.78/0.81      | **29.84/0.81**   |
> | 200    (latent)   | 27.98/0.78  | 29.04/0.80  | 29.82/0.81      | 29.82/0.81       |
>
> Our observations:
>
> * With more noise, more latent optimization is beneficial. Latent updates help enforce global structural consistency when measurements are noisy.
> * With less noise, more pixel optimization is is beneficial. Pixel refinements help sharpen details when data fidelity is reliable.
> * Early stopping prevents overfitting to noise, but at the cost of producing blurrier reconstructions with lower PSNR/SSIM.
> * The highest PSNR/SSIM occurs at the onset of overfitting, where reconstructions begin to “explain” measurement noise (Figure 14). It shows that external-consistency–based solvers can be sensitive and require careful balancing.
>
> We agree that adaptive iteration control, such as early stopping based on estimated noise level or timestep, could solve the artifact problem  but with lower reconstruction quality.
>
> > Including some GAN-based methods will strengthen this paper. If authors have time, I would also encourage trying some Gaussian splat based methods as these methods are gaining more popularity recently.
>
> Gaussian splatting is indeed an emerging and promising representation for 3D scenes. In the revised paper, we added a basic comparison using R2Gaussian (Tables 2, 10, 11; Figures 18 and 19). Overall, Gaussian splatting performs well in the noiseless medical setting, but its performance degrades in noisier scenarios. On the industrial dataset—where fine structures are abundant and noise levels are higher—the performance is noticeably weaker. We also note that Gaussian splatting pipelines typically rely on multiple custom CUDA extensions, making installation and reproducibility more challenging. We plan to further investigate integrating Gaussian-splatting techniques into our framework in a more accessible and standardized manner.
>
> Regarding GAN-based methods, we note that recent CT reconstruction literature has largely shifted away from adversarial priors due to issues like training instability and mode collapse. Multiple works have shown that diffusion models outperform GANs both for CT reconstruction specifically [1] and for inverse problems more broadly [2]. Incorporating GAN baselines would also require building a full adversarial training pipeline, which is beyond the scope of our benchmark and orthogonal to our goals. For these reasons, we discuss GANs conceptually but do not include new GAN experiments.

---

> ### Author Response · Authors · 2025-11-24
>
> > If authors have time, I am also interested in the performance of flow-based methods, such as FlowDPS or so on.
>
> Flow-based methods such as FlowDPS rely on large pretrained flow models (e.g., SD3-Flow), but such pretrained flow models do not currently exist for CT images. Our experiments on adapting pretrained natural image autoencoders to CT (Appendix A.11) indicate that fine tuning from natural images to CT is non-trivial, especially given the limited scale and low diversity of typical CT datasets. This suggests that adapting or pretraining a full flow model for CT would itself require substantial research effort.
>
> Training a flow model from scratch for CT would also be computationally demanding, and unlike diffusion, a pretrained flow model cannot be reused as a shared backbone across existing diffusion-based solvers in our benchmark. While we agree that flow-based approaches are an exciting and emerging direction, we consider them better suited for a dedicated follow-up study. We now clarify this point in the revised manuscript and explicitly mention flow-based methods in the future work section (line 501-512).

---

> > ### Comment · Reviewer_MUjs · 2025-11-25
> >
> > The author resolved my confusions and I raised my score to 8.

---

### Author Response · Authors · 2025-12-03
**Author Summary of Rebuttal and Clarifications**

We thank all reviewers for their time, effort, and constructive engagement. Below we summarize the discussion and how the concerns have been addressed.

Reviewer MUjs recognized the contributions of this work from the initial review: **a first comprehensive benchmark of current SOTA CT reconstruction methods**, **well-maintained and easy-to-use codebase**, **benchmark on different datasets is comprehensive** and **insightful analysis**. During the discussion (prior to the leakage incident), the reviewer explicitly stated that **the confusions are resolved and increased the score from 6 to 8**.

Reviewer KS57 highlighted that the benchmark includes  **comprehensive and impressive baselines** and provides a **unified framework for DM-based inverse solvers**. He raised several in-depth questions but emphasized that **the paper remains a valuable and well-structured contribution**, explicitly adding that he **would consider increasing my score** and **genuinely hope this line of inquiry will continue, as it has strong potential to advance CT reconstruction research**.

The remaining reviewer L1Ta raised concerns primarily centered on algorithmic novelty (**"not any new design of the model"**). We clarify that our submission is a **benchmark paper**, where novelty lies in **benchmark design**, **dataset contributions**, **unified implementations**, and **systematic evaluations and insights**. These contributions were recognized and appreciated by the other reviewers.

--------

### Contributions of this benchmark paper:

* Extensive and systematic benchmark on CT reconstruction
* Release of a new **high-resolution synchrotron CT dataset**, well suited for diffusion-model benchmarking
* Unified implementations and taxonomy of **10 diffusion methods**
* Multiple empirical insights, including:
  * the tradeoff between prior and data consistency,
  * external optimization strategies for latent diffusion methods,
  * the impact of diffusion training stages on CT reconstruction,
  * the exploration on potential value misalignment in values range between diffusion training and physical values in real data,
  * fine tuning existing pretrained autoencoders of RGB images for CT,
  * and more.

We believe the paper is correctly positioned as a benchmark paper and aligns well with expectations for the Datasets and Benchmarks track.

---
### Actions taken during the discussion phase

Following reviewer suggestions, we conducted additional experiments and expanded analyses. Key inquiries and corresponding actions include:

* Comparison to Gaussian Splatting and DDS -> both baselines included
* Diversity of experimental settings -> added limited-angle configuration
* Fine tuning pretrained SDXL/SD3 autoencoders -> added dedicated experiment section
* early stopping by noise level for pixel/latent consistency optimization -> added ablation for different noise levels
* Evaluation beyond PSNR/SSIM for medical CT -> added downstream segmentation task
* Effect of training stages on reconstruction -> added experiment comparing early/mid/final training stages
* Clarifications on sparse-angle noiseless reconstruction, scale of training data, real-data usage, and generalization
* Expanded discussion, limitations, and future-work sections.

---

### Remaining point

The only remaining concern is the question of algorithmic novelty. We believe this reflects a misunderstanding of the intended **scope of a benchmark submission** rather than a limitation of our work. Benchmark papers are fully within the scope of the ICLR, and the core contributions such as dataset, unified framework, systematic benchmark, and empirical insights, were clearly recognized and appreciated by the majority of reviewers.

---

We emphasize that **our goal is not to propose a new reconstruction algorithm**, but to provide **the first systematic benchmark for diffusion models in CT** (as stated in the initial submission). We believe the work fits the Datasets and Benchmarks track and can help the community with understanding and advancement of diffusion-based CT reconstruction methods. The released real-world high-resolution synchrotron dataset with straight forward geometry further supports future (diffusion-based) method development.

We sincerely thank the AC for the consideration and the effort invested in this decision process!

Best regards,

Authors

---

### Meta-Review · Area_Chair_A255 · 2026-01-14

**Summary:**

Reviewers generally indicate that:
1. The provided dataset is novel and constitutes a real world, high resolution CT benchmark that can be very useful benchmarking methods for solving medical inverse problems.
2. The provided benchmarks for different diffusion models is comprehensive and the performance analysis useful.
3. The benchmark miss other machine learning methods for solving inverse problems as well classical methods for CT reconstruction.
4. Hyper-parameter settings for different types of diffusion models are not fully explored.
5. The provided code base is well-written, easy to use, and provides a unified framework for ten different diffusion models.

**Reviewer Concerns:**

The authors have significantly expanded their methodology adding new methods, experimental settings, hyper parameters options, and analysis and discussions. The main concern of one reviewer is lack of novelty of the algorithms presented. However the goal is this work is not to present a new algorithm and rather introduce a new benchmark to the inverse problems community. In that goal, the work has succeeded.

**Reviewer Scores:**

One of the reviewers has not understood that algorithmic novelty is not the only measure which gives value to scientific work. I believe the score from that review does not accurately judge quality of the work.

---

### Decision · Program_Chairs · 2026-01-26

Accept (Poster)